# Geometry-aware Distance Measure for Diverse Hierarchical Structures in Hyperbolic Spaces

## Abstract

Learning in hyperbolic spaces has gained increasing attention due to the superior capability of modeling hierarchical structures. Existing hyperbolic learning methods use a fixed distance measure that assumes a uniform hierarchical structure across all data points. However, this assumption does not always hold in real-world scenarios, considering the diversity of the hierarchical structures of data. This work proposes to learn geometry aware distance measures that dynamically adjust to accommodate diverse hierarchical structures in hyperbolic spaces. We derive geometry aware distance measures by generating projections and curvatures for each pair of samples, which maps each pair to a suitable hyperbolic space. We introduce a revised low-rank decomposition scheme and a hard-pair mining mechanism to reduce the computational cost incurred by the pairwise generation without compromising accuracy. Moreover, we derive an upper bound of the low-rank approximation error via Talagrand concentration inequality to guarantee the effectiveness of our low-rank decomposition scheme. Theoretical analysis and experiments on standard image classification and few-shot learning tasks affirm the effectiveness of our method in refining hyperbolic learning through our geometry aware distance measures.

## 1 Introduction

The hyperbolic space is defined as a smooth Riemannian manifold with constant negative curvature. A notable property of Hyperbolic spaces lies in the exponential growth of a ball's volume relative to its radius, mirroring the exponential increase in the volume of hierarchical data with depth. Such a property enables the hyperbolic space to serve as a continuous analogous to trees (Sala et al., 2018; Balazevic et al., 2019), enabling to model hierarchical data with minimal distortion(Sarkar, 2011). The use of hyperbolic spaces for data embedding has shown to be superior in representing hierarchical structures across various applications such as classification (Gao et al., 2021; Zhang et al., 2022; Gao et al., 2022), clustering (Lin et al., 2022; 2023a), retrieval (Ermolov et al., 2022), segmentation (Hsu et al., 2021b; Atigh et al., 2022a; Chen et al., 2022), multi-modal (Hong et al., 2023a; Long & van Noord, 2023) and 3D vision (Hsu et al., 2021a; Montanaro et al., 2022; Leng et al., 2023; Lin et al., 2023b).

Existing hyperbolic learning methods usually deploy a fixed distance measure, *i.e.*, geodesic distance, to assess the similarities between data points, based on the expectation that the geodesic can effectively reflect the connecting paths between two nodes on the corresponding graph/tree(Behrstock et al., 2019). Employing a fixed distance measure implicitly includes the assumption of having a uniform hierarchical structure across all data points(Nickel & Kiela, 2018; Behrstock et al., 2017; 2019). However, this assumption does not always hold in real-world scenarios, as the hierarchical structures between data are diverse and complex. Thus, using a fixed distance measure in real-world scenarios may cause data distortion for diverse hierarchical structures, leading to sub-optimal performance.

For example, as shown in Figure 1(a), the "dog-wolf" data pair has a simple semantic hierarchical structure, while the "aircraft carriers-school bus" data pair has a more complex semantic hierarchical structure. In this case, a fixed distance measure fails to accurately represent the two different hierarchical relationships, unable to pass through the respective common ancestors, leading to a misalignment between distance and hierarchical structures, as detailed in Figure 1(b). Employing adaptive distance measures on data with diverse hierarchical structures seems a natural choice, which

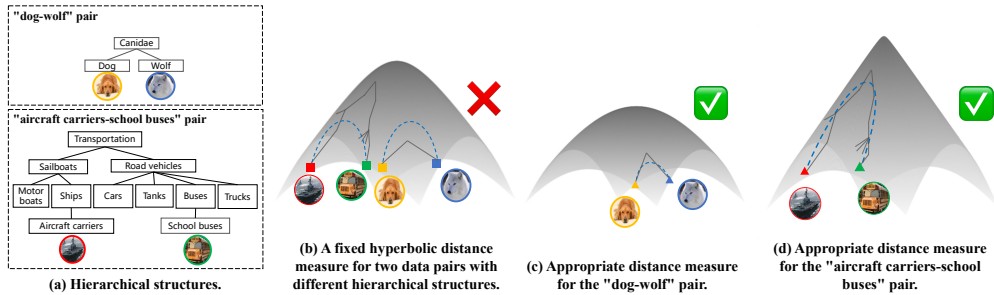

(a) Hierarchical structures.

(b) A fixed hyperbolic distance measure for two data pairs with different hierarchical structures.

(c) Appropriate distance measure for the "dog-wolf" pair.

(d) Appropriate distance measure for the "aircraft carriers-school buses" pair.

Figure 1: Modeling data with diverse hierarchical structures will benefit from adaptive distance measures. (a) The "dog-wolf" data pair has a simple semantic hierarchical structure, while the "aircraft carriers-school bus" data pair has a more complex structure. (b) A fixed distance measure fails to conform to the hierarchical structures of both data pairs simultaneously. (c) Lower $|c|$ (smaller space volume) and shorter distance are suitable for the simple structure. (d) Higher $|c|$ (larger space volume) and larger distances are suitable for the complex structure. The dashed lines represent geodesics in hyperbolic space. The solid lines represent the semantic subtrees. The color of the image border represents the category.

could take the complexity of the hierarchical structures into consideration and dynamically fit the diverse hierarchical structures, as shown in Figure 1(c) and (d).

In this paper, we propose to learn geometry-aware distance measures that automatically adapt to diverse hierarchical structures in hyperbolic spaces. Our main idea is to learn to generate adaptive projection and curvature for each pair of samples in the hyperbolic spaces, conforming to the hierarchical relationship between any two data points. In doing so, we design a curvature generator to produce adaptive curvatures for different data pairs and a projection matrix generator to map data pairs from the original hyperbolic space to an adaptive hyperbolic space with the new curvature. By applying adaptive projections and curvatures to the geodesic distance function for each sample pair, we obtain geometry-aware distance measures.

Two challenges need to be solved in learning to generate geometry-aware distance measures in the hyperbolic space: **(1)** Given that unique projections and curvatures are required for every data pair, reducing the computational cost becomes a critical consideration. **(2)** As a variety of hierarchical structures are encountered during the optimization process, the continuity of the optimization direction or trajectory for the pairwise level learning and training stability is not easily maintained. To address the first challenge, we introduce a low-rank decomposition scheme and a hard-pair mining mechanism. The former reduces computational complexity via low-rank approximation, and the latter eliminates easy samples, *i.e.*, we only generate adaptive distances for the remaining challenging ones. For the second challenge, we show that by incorporating residual connections into the projection matrix during the generation process, training stability can be well maintained. Experimental results in standard image classification and few-shot learning tasks confirm the effectiveness of our method in refining hyperbolic learning through geometry-aware distance measures.

The primary contributions of our work can be summarized as follows:

- We propose to learn to generate adaptive hyperbolic distances, enabling a more nuanced representation of diverse hierarchical structures inherent in data.

- We introduce a low-rank decomposition scheme and a hard-pair mining mechanism to significantly reduce computational costs without compromising accuracy.

- Theoretically, we prove that with a high probability, the low-rank decomposition in hyperbolic spaces yields small errors relative to the original full-rank matrix.

## 2 RELATED WORK

### 2.1 ADAPTIVE DEEP METRIC LEARNING

Adaptive metric learning is a pivotal machine learning technique that adapts embeddings or distance measures to handle data diversity (Li et al., 2021; 2019; Yoon et al., 2020; Liu & Wang, 2021). Adaptive embeddings are crafted through flexible prototypes (Li et al., 2021), tailored discriminative features (Li et al., 2019; Yoon et al., 2020), or episode-specific learning (Liu & Wang, 2021). Adaptive distance measurements range from task-specific metric spaces (Oreshkin et al., 2018; Qiao et al., 2019; Zhou et al., 2023) to optimized dynamic classifiers using subspaces (Simon et al., 2020). Recent advances include neighborhood-adaptive metric learning (Song et al., 2022; Li et al., 2022), yet these methods often rely on Euclidean space, limiting their effectiveness on data with intricate hierarchical structures. Different from these Euclidean-based methods, we propose to learn geometry-aware distance measures in hyperbolic spaces, which can exploit the inherent hierarchical structures of the data.

### 2.2 HYPERBOLIC GEOMETRY

Hyperbolic geometry has shown superior performances in many applications due to their capabilities in modeling data with hierarchical structures. Research on hyperbolic geometry can be divided into three categories. Methods of the first category opt for modeling several applications on hyperbolic spaces, such as medical image recognition (Yu et al., 2022), action recognition (Long et al., 2020), audio-visual learning (Hong et al., 2023b), image segmentation (Atigh et al., 2022b), anomaly detection (Li et al., 2024), and 3D Visual Grounding (Wang et al., 2024) . Methods of the second category work on extending convincing neural architectures from Euclidean spaces to hyperbolic spaces, such as convolutional network (Shimizu et al., 2021) and graph network (Dai et al., 2021). Methods of the third category focus on extending learning paradigms from Euclidean spaces to hyperbolic manifolds, such as contrastive learning (Ge et al., 2023), self-supervised learning (Franco et al., 2023), and metric learning (Yan et al., 2021). Different from these methods, we propose to learn geometry-aware hyperbolic distance measures for each data pair to match its unique hierarchical structure.

### 2.3 CURVATURE LEARNING

Previous studies (Gu et al., 2019) have established that selecting the appropriate curvature is crucial for effective hyperbolic learning, as it significantly impacts the quality of the learned representations. Research on curvatures has continually advanced, encompassing the application of constant (Bachmann et al., 2020) in graph neural networks, as well as the exploration of curvature learning methods (Yang et al., 2023). Gu et al. (Gu et al., 2019) address embedding diverse hierarchical data by using a product manifold that combines multiple spaces with heterogeneous curvature. Inspired by these methods, our model takes it a step further by dynamically adjusting curvature for pair-wise distance measures. Our method stands out by dynamically adapting curvature for each pair during inference, based on their specific characteristics.

## 3 MATHEMATICAL PRELIMINARIES

**Notations.** In the following sections, $\mathbb{R}^n$ denotes $n$-dimensional Euclidean space and $\|\cdot\|$ denotes the Euclidean norm. The vectors are denoted by lower-case letters, such as $\boldsymbol{x}$ and $\boldsymbol{y}$. The matrices are denoted by upper-case letters, such as $\boldsymbol{M}$. The Poincaré ball model of an $n$-dimensional hyperbolic space with curvature $c$ ($c < 0$) is defined as a Riemannian manifold $\left(\mathbb{B}_c^n, h_c^B\right)$, where $\mathbb{B}_c^n = \{\boldsymbol{x} \in \mathbb{R}^n : -c\|\boldsymbol{x}\| < 1, c < 0\}$ is the open ball with radius $1/\sqrt{|c|}$ and $h_c^B$ is the Riemannian metric. The tangent space at $\boldsymbol{x} \in \mathbb{B}_c^n$, a Euclidean space, is denoted by $T_{\boldsymbol{x}}\mathbb{B}_c^n$. We use the Möbius gyrovector space (Ungar, 2001) that provides operations for hyperbolic learning and several used operations are shown as follows. More details regarding the Poincaré ball model and its properties are provided in the **Appendix** A.

**Addition.** For a pair $\boldsymbol{x}, \boldsymbol{y} \in \mathbb{B}_c^n$, the Möbius addition is defined as

$$\boldsymbol{x} \oplus_c \boldsymbol{y} = \frac{\left(1 - 2c\langle\boldsymbol{x}, \boldsymbol{y}\rangle - c\|\boldsymbol{y}\|^2\right)\boldsymbol{x} + \left(1 + c\|\boldsymbol{x}\|^2\right)\boldsymbol{y}}{1 - 2c\langle\boldsymbol{x}, \boldsymbol{y}\rangle + c^2\|\boldsymbol{x}\|^2\|\boldsymbol{y}\|^2}. \tag{1}$$

**Distance function.** The geodesic distance $d_c(\cdot, \cdot)$ between two points $\boldsymbol{x}, \boldsymbol{y} \in \mathbb{B}_c^n$ can be obtained as

$$d_c(\boldsymbol{x}, \boldsymbol{y}) = \frac{2}{\sqrt{c}} \operatorname{arctanh}\left(\sqrt{c}\,\|-\boldsymbol{x} \oplus_c \boldsymbol{y}\|\right). \tag{2}$$

**Möbius matrix multiplication.** In the Gyrovector space, the Möbius matrix multiplication $\otimes_c$ for the matrix $\boldsymbol{M} \in \mathbb{B}^{n \times n}$ and vector $\boldsymbol{x} \in \mathbb{B}^n$ is defined as

$$\boldsymbol{M} \otimes_c \boldsymbol{x} = \frac{1}{\sqrt{|c|}}\tanh\left(\frac{\|\boldsymbol{M}\boldsymbol{x}\|}{\|\boldsymbol{x}\|}\operatorname{arctanh}(\sqrt{|c|}\|\boldsymbol{x}\|)\right)\frac{\boldsymbol{M}\boldsymbol{x}}{\|\boldsymbol{M}\boldsymbol{x}\|}. \tag{3}$$

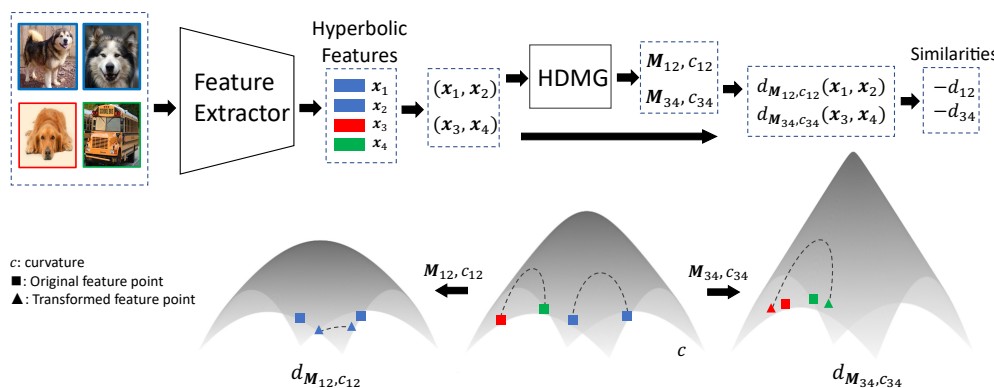

Figure 2: Overview of the proposed method. Two data pairs(one positive and one negative) are encoded with the feature extractor, while the color of the embeddings indicates the class of each data point. "HDMG" indicates the hyperbolic distance measure generator, which generates the distance measures according to the data pair. $d_{\boldsymbol{M}_{12}, c_{12}}$ and $d_{\boldsymbol{M}_{34}, c_{34}}$ are computed from Eq. (5). Under the transformed distance measure, the positive samples are pulled closer, while the negative samples are pushed further away.

## 4 METHOD

### 4.1 ANALYSES

We first analyze the relationship between the distance measure and the complexity of hierarchical structures in the hyperbolic space. We define the complexity of the hierarchical structure of the data pair $\boldsymbol{x}_i$ and $\boldsymbol{x}_j$ as

$$C(\boldsymbol{x}_i, \boldsymbol{x}_j) = P(\boldsymbol{x}_i \to \boldsymbol{o}) + P(\boldsymbol{x}_j \to \boldsymbol{o}), \tag{4}$$

where $\boldsymbol{o}$ is the origin, and $P(\cdot \to \boldsymbol{o})$ is the connectivity from $\boldsymbol{x}$ to $\boldsymbol{o}$, measured by the connected graph distance(Balbuena et al., 1996).

We find that the suitable distance measure varies to be faithful to different hierarchical structures, as shown in the following proposition.

**Proposition 4.1.** *In a hyperbolic space $\mathbb{B}_c^n$, a steeper geodesics $d(\boldsymbol{x}_i, \boldsymbol{x}_j)$ (e.g., a large curvature c) can better conform to complex hierarchical structures with higher $C(\boldsymbol{x}_i, \boldsymbol{x}_j)$ in equation 4, and vice versa.*

The proof can be found in Appendix B.2, which motivates us to design geometry-aware distance measures to match diverse hierarchical structures in practical data.

## 4.2 FORMULATION

Given the hyperbolic features $\{\boldsymbol{x}\}$, our method produces an adaptive projection matrix $\boldsymbol{M}_{ij} \in \mathbb{B}_{c_{ij}}^{n \times n}$ and curvature $c_{ij}$ for one pair of features $(\boldsymbol{x}_i, \boldsymbol{x}_j)$. For ease of exposition, we will use $\boldsymbol{M}$ to denote the $\boldsymbol{M}_{ij}$ and use $c$ to refer to $c_{ij}$. In this case, the distance between $(\boldsymbol{x}_i, \boldsymbol{x}_j)$ is

$$d_{\boldsymbol{M},c}(\boldsymbol{x}_i, \boldsymbol{x}_j) = d_c(\boldsymbol{M} \otimes_c \boldsymbol{x}_i, \boldsymbol{M} \otimes_c \boldsymbol{x}_j), \tag{5}$$

where $d_c(\cdot, \cdot)$ is the Poincaré geodesic distance function (Eq. (2)).

The goal of our method is to train the matrix generator $g_t$ and curvature generator $g_c$, through which positive pairs are closer and negative pairs are pushed farther apart. $\boldsymbol{M}$ is produced by the matrix generator $g_t$: $\boldsymbol{M} = g_t(\boldsymbol{x}_i, \boldsymbol{x}_j)$, and $c$ is generated by the curvature generator $g_c$: $c = g_c(\boldsymbol{x}_i, \boldsymbol{x}_j)$.

In practical applications, generating projection matrices and curvatures for all pairs would result in significant computational costs. To reduce the computation overhead, we introduce a hard-pair mining mechanism to filter out the hard pairs $\mathcal{H} = \text{HPM}(D_{train})$, where $\mathcal{H}$ denotes the hard cases, $D_{train}$ is the training set and $\text{HPM}(\cdot)$ denotes the hard-pair mining. Our method only generates projection matrices and curvatures for pairs in $\mathcal{H}$. In the following section, we will introduce $g_t$, $g_c$, and hard-pair mining in detail.

## 4.3 GEOMETRY-AWARE HYPERBOLIC DISTANCE MEASURES

The schematic overview of our method is depicted in Figure 2. We begin by using a feature extractor to obtain hyperbolic features from image pairs. We then produce an adaptive projection matrix and curvature for each pair of images through which we are able to conform to the hierarchical structures between the images.

### 4.3.1 PROJECTION MATRIX GENERATOR

The distance measure generator takes a pair of feature points $\boldsymbol{x}_i$ and $\boldsymbol{x}_j$ as inputs and provides a transformation matrix $\boldsymbol{M}$ as output. In this paper, we utilize an adaptive matrix generator to produce the projection matrix $\boldsymbol{M}$. Excessive changes of $\boldsymbol{M}$ will result in instability of training. Instead of directly generating the matrix $\boldsymbol{M}$, we learn a residual $\boldsymbol{M}^{res}$ between the original distance measure (in Eq. (2)) and the geometry aware distance measure (in Eq. (5)) to tackle this issue. Therefore, the projection matrix $\boldsymbol{M}$ is computed by $\boldsymbol{M} = \boldsymbol{I} + \boldsymbol{M}^{res}$. From Equations (5), it can be seen that the Poincaré geodesic distance is a special case of our distance measure when $\boldsymbol{M}^{res}$ is a zero matrix. Considering that when the dimension of the embeddings $n$ is relatively large, generating and operating on $\boldsymbol{M}^{res} \in \mathbb{B}^{n \times n}$ can cause substantial computational overhead. Therefore, we decompose $\boldsymbol{M}^{res}$ into the product of two low-rank matrices: $\boldsymbol{M}^{res} = \boldsymbol{M}_a^{res} \boldsymbol{M}_b^{res\top}$, where $\boldsymbol{M}_a^{res} \in \mathbb{B}^{n \times k}$ and $\boldsymbol{M}_b^{res} \in \mathbb{B}^{n \times k}$ are computed by $\boldsymbol{M}_{ij,a}^{res} = f_a(\boldsymbol{x}_i, \boldsymbol{x}_j)$, $\boldsymbol{M}_{ij,b}^{res} = f_b(\boldsymbol{x}_i, \boldsymbol{x}_j)$, respectively, $f_{a/b}(\cdot, \cdot)$ are fully connected layers and $k$ is the rank(much smaller than $n$). Substituting the residual matrix of low-rank decomposition Substituting the residual matrix of low-rank decomposition into $M$, we can obtain that

$$\boldsymbol{M} = \boldsymbol{I} + \boldsymbol{M}_a^{res} \boldsymbol{M}_b^{res\top}. \tag{6}$$

**Low-rank approximation** Based on polynomial partitioning (developed by Larry Guth and Nets Katz when dealing with the Erdös' distinct distances problem (L. Guth, 2015) and Talagrand concentration inequality (Talagrand, 1995), we prove that the low-rank matrices $\boldsymbol{M}' = \boldsymbol{I} + \boldsymbol{M}_a^{res} \boldsymbol{M}_b^{res\top}$ can be well approximated to the full-rank matrix $\boldsymbol{M} = \boldsymbol{I} + \boldsymbol{M}^{res}$. We provide the upper bound of the approximation error, *i.e.*, $|\boldsymbol{M} \otimes \boldsymbol{x} - \boldsymbol{M}' \otimes \boldsymbol{x}|$, as well as the lower bound of the probability for this upper bound to hold, where $\otimes$ is möbius matrix multiplication (Eq. 3) with $c = -1$ and $|\cdot|$ represents absolute value calculation.

**Theorem 4.2.** *Suppose the variance of systematic error $\sigma^2 = \epsilon^2/n$ and $\epsilon$ is significantly smaller than (say $\sim \frac{1}{10}$ of) the mean of error, and the samples feature space distribute relatively continuously. Then with high probability ($1 - ce^{-k}$ for some constant $c > 0$ say for $k \sim n/10$ or $k \sim \sqrt{n}$), for any $\boldsymbol{x} \in \mathbb{R}^n$, the low rank (say $\leq k \leq n/10$ or $\sim \sqrt{n}$ with $n$ the dimension of features) approximation error $|\boldsymbol{M} \otimes \boldsymbol{x} - \boldsymbol{M}' \otimes \boldsymbol{x}|$ is bounded by $C\epsilon$ for some absolute constants $C > 0$.*

**Remark.** Theorem 4.2 encapsulates the lower bound of the probability of maintaining acceptable error bounds when substituting a full-rank matrix $\boldsymbol{M}$ with its low-rank counterpart $\boldsymbol{M}'$. Note that in

practice the dimension $n$ is taken to be hundreds say $512$ in most of our experiments. The theorem only claims that with high probability which goes to 1 as $k$ and $n$ grows, the low-rank approximation is bounded by the system error but not that it diminishes to none as the number of parameters grows. Also, the probability bound is not optimal. Detailed derivation can be found in the **Appendix** D.

### 4.3.2 CURVATURE GENERATOR

We use the factorized bilinear pooling(Yu et al., 2017) to produce suitable curvature for the pair of $\boldsymbol{x}_i$ and $\boldsymbol{x}_j$ since the expressive second-order information in the factorized bilinear pooling could benefit from discovering how curvature reflects the degree of warping in the hyperbolic space(Gao et al., 2021). More details can be found in **Appendix** C. Firstly, we used two fully connected layers $f_1$ and $f_2$ to process $\boldsymbol{x}_i$ and $\boldsymbol{x}_j$ separately: $\boldsymbol{x}'_i = f_1(\boldsymbol{x}_i), \ \ \boldsymbol{x}'_j = f_2(\boldsymbol{x}_j)$.

Then, the Hadamard product of the matrices was computed, producing the matrix $\boldsymbol{W} = \boldsymbol{x}'_1 \circ \boldsymbol{x}'_2$, where $\circ$ denotes the Hadamard product. $\boldsymbol{W}$ is subsequently transformed into curvature $c$ by using sum pooling. Finally, to limit $c$ within the range of $[0, 1]$, we applied a sigmoid layer($\sigma(\cdot)$):

$$c = \sigma(\text{sum\_pooling}(\boldsymbol{W})). \tag{7}$$

### 4.4 HARD-PAIR MINING IN HYPERBOLIC SPACE

In real-world scenarios, hierarchical complexity varies widely. Many samples with simple hierarchies can be described well by a fixed hyperbolic distance measure, while creating geometry-aware measures may incur high computational costs. To tackle this, we propose a hard-pair mining mechanism to identify difficult pairs without extra parameters. Our approach exclusively generates distance measures for these pairs.

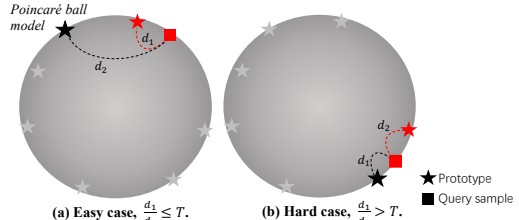

(a) Easy case, $\frac{d_1}{d_2} \le T.$  (b) Hard case, $\frac{d_1}{d_2} > T.$

For the classification task, we compute the hyperbolic distance between the query sample and all prototypes in the Poincaré ball model. We then identify the two closest prototypes, with distances $d_1$ (nearest) and $d_2$ (second nearest). By calculating the ratio $d_1/d_2$, we determine the proximity of the sample to the classification boundary. A ratio closer to 1 indicates a harder case for accurate classification. For a feature set $\mathcal{F}_q$ of the query set and prototypes $\mathcal{P}$, the hard-case mining process can be represented as

Figure 3: Hard-pair mining mechanism in hyperbolic space. The threshold $T \in [0, 1]$ is a margin hyperparameter. Embedding classes are color-coded, and Poincaré geodesic distances are shown as dashed lines. The distances from the query sample to the closest and second closest prototypes are $d_1$ and $d_2$, respectively. Hard cases, difficult to classify, are defined by $d_1/d_2 > T$.

$$\mathcal{H} = \{\boldsymbol{x} \mid \boldsymbol{x} \in \mathcal{F}_q, \frac{d(\boldsymbol{x}, \boldsymbol{p}_1)}{d(\boldsymbol{x}, \boldsymbol{p}_2)} > T\}, \quad (8)$$

where $d(\cdot, \cdot)$ is the Poincaré geodesic distance function in Eq. (2), while $\boldsymbol{p}_1 = \mathrm{argmin}_{\boldsymbol{p} \in \mathcal{P}} d(\boldsymbol{x}, \boldsymbol{p})$ and $\boldsymbol{p}_2 = \mathrm{argmin}_{\boldsymbol{p} \in \mathcal{P} - \boldsymbol{p}_1} d(\boldsymbol{x}, \boldsymbol{p})$. $T \in [0, 1]$ is a margin hyperparameter. Figure 3 shows the hard-pair mining mechanism applied in the Poincaré ball model.

### 4.5 TRAINING PROCESS

The goal of this work is to learn the distance measure generator, including $g_t$ and $g_c$. During the training process, we partition the training set $\mathcal{D}$ into the support set $\mathcal{D}_s$ and the query set $\mathcal{D}_q$. We extract features $\mathcal{F}_s$ and $\mathcal{F}_d$ from $\mathcal{D}_s$ and $\mathcal{D}_q$, respectively, using the backbone network with exponential map. We denote the set of prototypes of $\mathcal{P}$. The hard cases $\mathcal{H}$ are selected from $\mathcal{F}_q$ via hard-pair mining in Eq. (8). Then we generate the adaptive matrices and curvatures using $g_t$ and $g_c$ via Eq. (6) from the pair set $\mathcal{H} \times \mathcal{P}$, and compute the adaptive distance as the logits via Eq. (5). We update $g_t$ and $g_c$ by minimizing the cross entropy loss of $\mathcal{D}_q$. The pseudo-code of training is summarized in Algorithm 1.

---

**Algorithm 1** Training process of our method.

---

**Require:** Training set $\mathcal{D}$.
**Ensure:** The updated metric generator $g_t$ and $g_c$.
 1: **while** Not converged **do**
 2:     Randomly sample the support set $\mathcal{D}_s$ and query set $\mathcal{D}_q$ from $\mathcal{D}$.
 3:     Extract features $F_s$ and $F_d$ from $\mathcal{D}_s$ and $\mathcal{D}_q$, respectively.
 4:     Calculate the Einstein mid-point as prototypes $\mathcal{P}$ from $F_s$.
 5:     Select the hard cases $\mathcal{H}$ from $\mathcal{F}_q$ via Eq. (8).
 6:     Generate the adaptive matrices and curvatures via $g_t$ and $g_c$ using Eq. (6) from the pair set $\mathcal{H} \times \mathcal{P}$, and compute the distance as the logits via Eq. (5).
 7:     Compute the cross entropy loss, and update $g_t$ and $g_c$.
 8: **end while**

---

### 4.6 COMPLEXITY ANALYSIS

Low-rank distance measures reduce computational costs in generating projection matrices. Our method produces two low-rank matrices ($n \times k$) with a cost of $O(n^2 k)$, compared to $O(n^3)$ for directly generating an $n \times n$ matrix, where $k \ll n$.

The time complexities for our method's components are $O(pq)$ for hard-pair mining, $O(n^2)$ for curvature generation, and $O(n^2 k)$ for projection matrix generation. Here, $p$ is the number of classes, $q$ the number of queries, $n$ the dimensionality, and $k$ the rank. The overall time complexity is $O(n^2(k+1) + pq) = O(n^2(k+1))$, as $p$ and $q$ are much smaller than $n$.

## 5 EXPERIMENT

We evaluate our method on standard classification and few-shot learning tasks. We use common backbone networks with the exponential map as the feature extractor, then we apply the hard-pair mining mechanism to select the hard cases, and finally generate the distance measures for the hard cases. The rank is set to 16 for all settings in Section 5.1 and Section 5.3. More experiments (visualization and ablation) and model setups can be found in the **Appendix** E and **Appendix** F.

### 5.1 STANDARD CLASSIFICATION

We conduct experiments on three datasets: MNIST (LeCun & Cortes, 2010), CIFAR10 (Krizhevsky et al., 2009) and CIFAR100 (Krizhevsky et al., 2009) datasets. Full details of the datasets, implementation and pretraining are described in the **Appendix** F. We learn class prototypes and classify by calculating distances between these prototypes and test set features. Table 1 compares our method with existing hy-

Table 1: Accuracy (%) comparisons with existing hyperbolic learning methods on the MNIST, CIFAR10 and CIFAR100 datasets.

| Method | MNIST | CIFAR10 | CIFAR100 |
|---|---|---|---|
| Hyp-Optim | 94.42 | 88.82 | 72.26 |
| HNN++ | 95.01 | 91.22 | 73.65 |
| Hyp-ProtoNet | 93.53 | 93.30 | 73.83 |
| Ours | **96.56** | **94.75** | **75.61** |

perbolic learning methods. On MNIST, CIFAR10, and CIFAR100, our method improves by 3.03%, 1.45%, and 1.78% over Hyp-ProtoNet (Khrulkov et al., 2020), and by 2.14%, 5.93%, and 3.35% over Hyp-Optim (Ganea et al., 2018). Compared to HNN++ (Shimizu et al., 2021), our method achieves 1.55%, 3.53%, and 1.9% higher accuracy. These results demonstrate that our adaptive distance measures outperform existing hyperbolic learning methods by better matching inherent hierarchical structures.

### 5.2 HIERARCHICAL CLASSIFICATION

We utilize the CIFAR100 dataset and its 5-level hierarchical annotations from (Wang et al., 2023) (detailed annotations can be found in the **Appendix** F.2.1). We report our accuracies on the 5 hierarchical levels using Resnet50 and Resnet101, as shown in Table 2. Compared with using a fixed distance measure (denoted as 'fixed' in Table 2), our adaptive distance measure has better performance on all hierarchical levels, indicating that classes belonging to the same parent node are closely grouped after our projection. Then results demonstrate that our model can effectively capture

Table 2: Hierarchical accuracy (%) of the fixed distance measure vs. our method on the CIFAR-100 dataset. Levels 0 to 4 (coarse-to-fine) represent test results at different levels of annotation.

| Method | Level 0 | Level 1 | Level 2 | Level 3 | Level 4 |
|---|---|---|---|---|---|
| Resnet50+fixed | 95.62 | 90.65 | 88.68 | 86.30 | 78.49 |
| Resnet50+ours | **96.50** | **91.88** | **90.22** | **88.11** | **81.19** |
| Resnet101+fixed | 95.95 | 91.51 | 90.08 | 87.87 | 80.97 |
| Resnet101+ours | **97.88** | **93.68** | **92.27** | **90.13** | **83.44** |

the implicit hierarchical structure within the data. We also visualize the embedding distribution at each level, details can be found in **Appendix** E.7.1.

## 5.3 FEW-SHOT LEARNING

We conducted experiments on two popular few-shot learning datasets: mini-ImageNet (Vinyals et al., 2016) and tiered-ImageNet (Ren et al., 2018). Full details of the datasets, implementation and pretraining are described in the **Appendix** F. We compare our method with the hyperbolic methods, the metric-based Hyp-ProtoNet (Khrulkov et al., 2020) and the optimization-based Hyp-Kernel (Fang et al., 2021), C-HNN (Guo et al., 2022) and CurAMl(Gao et al., 2023), as shown in Table 3. Note that Hyp-ProtoNet (Khrulkov et al., 2020) is a fixed metric-based hyperbolic few-shot learning method, compared with it, our method is 5.28% 1-shot and 5.05% 5-shot higher than it, suggesting that our method generates better distance measures for matching the inherent hierarchical structures of data. We also compare our method with the popular Euclidean optimization-based (Finn et al., 2017; Baik et al., 2020; 2021; Gao et al., 2021; Sun & Gao, 2023) and Euclidean metric-based (Snell et al., 2017; Vinyals et al., 2016; Lee et al., 2019; Lu et al., 2021; Simon et al., 2020; Oreshkin et al., 2018; Li et al., 2020; Yoon et al., 2020; Khrulkov et al., 2020) few-shot learning methods. Our method improves the optimization-based methods in the Euclidean space on both the 1-shot and the 5-shot tasks. Compared with the fixed Euclidean metric-based methods(Snell et al., 2017; Simon et al., 2020; Huang et al., 2021) , our method brings more than 1% improvements on the 1-shot task and 2% on the 5-shot task. Compared with the adaptive metric-based methods in the Euclidean space, such as TADAM (Oreshkin et al., 2018) and XtarNet (Yoon et al., 2020), our method exceeds them in both 1-shot and 5-shot accuracy. The main reason is that performing metric learning in the hyperbolic space preserves the hierarchical structures of data and avoids undesirable data distortion.

Table 3: Accuracy (%) comparisons with popular few-shot learning methods on the mini-ImageNet and tiered-ImageNet datasets. 'Optim' and 'Metric' mean the optimization-based and metric-based few-shot learning methods, respectively. 'Euc' and 'Hyp' mean the methods are performed in the Euclidean space and the Hyperbolic space, respectively. '*' indicates that results use ResNet-18 (He et al., 2016) as the backbone, while the others use ResNet-12 (He et al., 2016).

| Method | Space | Category | min-ImageNet | | tiered-ImageNet | |
|---|---|---|---|---|---|---|
| | | | 1-shot | 5-shot | 1-shot | 5-shot |
| MAML (Finn et al., 2017) | Euc | Optim | 51.03 ± 0.50 | 68.26 ± 0.47 | 58.58 ± 0.49 | 71.24 ± 0.43 |
| L2F (Baik et al., 2020) | Euc | Optim | 57.48 ± 0.49 | 74.68 ± 0.43 | 63.94 ± 0.84 | 77.61 ± 0.41 |
| MeTAL (Baik et al., 2021) | Euc | Optim | 59.64 ± 0.38 | 76.20 ± 0.19 | 63.89 ± 0.43 | 80.14 ± 0.40 |
| Meta-AdaM (Sun & Gao, 2023) | Euc | Optim | 59.89 ± 0.49 | 77.92 ± 0.43 | 65.31 ± 0.48 | 85.24 ± 0.35 |
| ProtoNet (Snell et al., 2017) | Euc | Fixed Metric | 56.52 ± 0.45 | 74.28 ± 0.20 | 53.51 ± 0.89 | 72.69 ± 0.74 |
| DSN (Simon et al., 2020) | Euc | Fixed Metric | 62.64 ± 0.66 | 78.83 ± 0.45 | 66.22 ± 0.75 | 82.79 ± 0.48 |
| LMPNet (Huang et al., 2021) | Euc | Fixed Metric | 62.74 ± 0.11 | 80.23 ± 0.52 | 70.21 ± 0.15 | 7945 ± 0.17 |
| TADAM (Oreshkin et al., 2018) | Euc | Adaptive Metric | 58.50 ± 0.30 | 76.70 ± 0.30 | - | - |
| XtarNet (Yoon et al., 2020) | Euc | Adaptive Metric | 55.28 ± 0.33 | 66.86 ± 0.31 | 61.37 ± 0.36* | 69.58 ± 0.32* |
| Hyp-Kernel (Fang et al., 2021) | Hyp | Optim | 61.04 ± 0.21* | 77.33 ± 0.15* | 57.78 ± 0.23* | 76.48 ± 0.18* |
| C-HNN (Guo et al., 2022) | Hyp | Optim | 53.01 ± 0.22 | 72.66 ± 0.15 | - | - |
| CurAML (Gao et al., 2023) | Hyp | Optim | 63.13 ± 0.41 | 81.04 ± 0.39 | 68.46 ± 0.56 | 83.84 ± 0.40 |
| Hyp-ProtoNet (Khrulkov et al., 2020) | Hyp | Fixed Metric | 59.47 ± 0.20* | 76.84 ± 0.14* | - | - |
| Ours | Hyp | Adaptive Metric | **64.75 ± 0.20** | **81.89 ± 0.15** | **72.59 ± 0.22** | **86.14 ± 0.16** |

## 5.4 ABLATION

### 5.4.1 EFFECTIVENESS OF THE HYPERBOLIC DISTANCE MEASURE GENERATOR

We evaluate the effectiveness of different components in our hyperbolic distance measure generator on the tiered-ImageNet dataset. We compare ours (iv) with the following three distinct experimental setups. (i) Fixed hyperbolic metric: We employ the Poincaré geodesic distance function, as defined in Eq. (2). (ii) Adaptive curvature only: We deactivate the projection matrices generator $g_t$ and solely

Table 4: 5-shot accuracy(%) and and 95 % confidence interval on tiered-ImageNet dataset. The hard-pair mining is deactivated.

| Metric | Fixed hyp metric | Ours w/o $g_t$ | Ours w/o $g_c$ | Ours |
|---|---|---|---|---|
| 5-shot acc(%) | 83.94 ± 0.16 | 84.67 ± 0.15 | 84.85 ± 0.16 | **86.10 ± 0.16** |

utilize the adaptive curvature generator $g_c$ with Eq. (2). (iii) Projection matrices only : We activate the $g_t$ but disable the $g_c$, setting the curvature to 0.5. Results are shown in Table 4.

In Table 4, the adaptive curvature generator $g_c$ provides a more discriminative feature space than the fixed curvature space, *i.e.*, (i) *vs.*(ii). As evidenced in rows (i) and (iii) of Table 4, our method benefits from the projection matrix generator $g_t$ and the geometry aware distance measures match better with the inherent hierarchical structures than the fixed distance measure.

### 5.4.2 EFFECTIVENESS OF THE RESIDUAL CONNECTION

We conduct experiments about the accuracy and training loss w/ and w/o the residual connection to verify its effectiveness on the CIFAR100 dataset. Using the residual solution brings 1.07% improvements (w/o res (74.54%) *vs.* w/ res(75.61%)). The loss curves in Figure 4 show that using the residual connection brings stable training process with faster convergence and smoother loss curves.

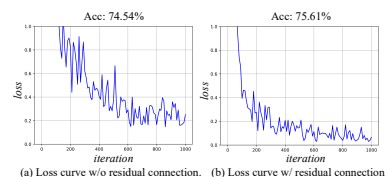

(a) Loss curve w/o residual connection. (b) Loss curve w/ residual connection.

Figure 4: Loss curves on the CIFAR100 dataset.

### 5.4.3 EFFECT OF THE RANK IN MATRIX DECOMPOSITION

We further explore the effect of rank in matrix decomposition on the mini-ImageNet dataset. Our low-rank decomposition decomposes the original $n \times n$ matrices (for ResNet-12 backbone, $n = 512$) multiplication into two $n \times k$ matrices multiplication, greatly reducing the computational complexity from $O(n^3)$ to $O(nk^2), k << n$. Here, we evaluate the value of $k$ in the range of $[4, 8, 16, 32, 64]$, and report the accuracy and memory cost. As shown in Table 5, the accuracy increased first and then decreased as the rank increased. As the rank increases, we retain more and more information, resulting in an increase in accuracy. However, when the rank becomes too large,

Table 5: 5-shot accuracy(%), memory cost(MB) and time cost(ms) per few shot learning task with different ranks. The memory cost of one inference process with the setting of 5w5s and 15 queries. We disable the hard-pair mining when testing the memory cost.

|  | Rank | 5-shot acc(%) | Mem(MB) | Time(ms) |
|---|---|---|---|---|
| **w/ ours** | 4 | 81.41 ± 0.14 | 98.06 | 5.02 |
|  | 8 | 81.53 ± 0.14 | 179.56 | 5.25 |
|  | 16 | 81.80 ± 0.14 | 352.86 | 6.10 |
|  | 32 | 81.48 ± 0.14 | 703.10 | 8.67 |
|  | 64 | 81.45 ± 0.14 | 1371.64 | 14.26 |
| **w/o ours** | 512 | 81.74 ± 0.14 | 7231.24 | 60.88 |

excessive information, including errors and noise, may be preserved, which can lead to overfitting and a decrease in accuracy. As rank increases, the number of model parameters and the computational cost both increase significantly. As shown in Table 5, when $k = 64$, the total memory consumption is nearly four times that of $k = 16$. Considering the trade-off between accuracy and computational cost, this paper selects $k = 16$.

### 5.4.4 EFFECTIVENESS OF THE HARD-PAIR MINING

Table 6: Effectivness of the hard-pair mining. Threshold is the $T = d_1/d_2$. The 'Percentage' represents the proportion of hard cases in the query set. The rest columns represent the 5-shot accuracy(%) of the easy cases with Eq. (2), hard cases with Eq. (2), hard cases with our methods, total query set with Eq. (2) and total query set with our method, respectively.

| Threshold | Percentage | Easy cases w/ Eq. (2) | Hard cases w/ Eq. (2) | Hard cases w/ ours | Total w/ Eq. (2) | Total w/ ours |
|---|---|---|---|---|---|---|
| 0.1 | 100% | - | 81.26 ± 0.14 | **81.61 ± 0.14** | 81.26 ± 0.14 | **81.61 ± 0.14** |
| 0.8 | 89% | 98.48 ± 0.14 | 78.97 ± 0.14 | **79.39 ± 0.14** | 81.18 ± 0.14 | **81.54 ± 0.14** |
| 0.9 | 57% | 97.07 ± 0.08 | 69.17 ± 0.17 | **69.91 ± 0.17** | 81.14 ± 0.14 | **81.53 ± 0.14** |
| 0.96 | 22% | 89.29 ± 0.11 | 53.52 ± 0.24 | **55.62 ± 0.25** | 81.38 ± 0.14 | **81.89 ± 0.14** |
| 0.98 | 14% | 86.51 ± 0.13 | 49.40 ± 0.35 | **52.41 ± 0.36** | 81.27 ± 0.14 | **81.66 ± 0.14** |
| 0.99 | 7% | 83.73 ± 0.23 | 45.92 ± 0.51 | **50.35 ± 0.53** | 81.21 ± 0.23 | **81.49 ± 0.23** |

We further evaluate the effectiveness of the hard-pair mining mechanism on the mini-ImageNet dataset, as shown in Table 6. We assess threshold values in the range of $[0.1, 0.8, 0.9, 0.96, 0.98, 0.99]$ and

report 5-shot accuracy using both the fixed distance measure (Eq. (2)) and the adaptive distance measure (Eq. (5)) for easy cases, hard cases, and the total query set. The results in Table 7 show that our mechanism effectively selects hard examples. At $T = 0.9$, 57% of samples are classified as hard. As the threshold increases, the proportion of hard cases decreases, reducing computational complexity by focusing on adaptive measures for hard cases. The mechanism also distinguishes effectively between easy and hard cases, with higher accuracy for the former. Our hyperbolic distance measure generator significantly improves classification accuracy for hard cases. As difficulty increases, the benefits of our method become more pronounced. For thresholds of 0.96, 0.98, and 0.99, accuracy improvements for hard cases are 2.1%, 3.01%, and 4%, respectively, compared to cases without our method.

Table 7: Percentage of hard pairs, 5-shot accuracy(%) and time per few-shot learning task on the mini-ImageNet dataset. HPM denotes the hard-pair mining.

|  | Hard-pair percentage(%) | Hard-pair percentage(%) | Time of HPM(ms) | Running time(ms) |
|---|---|---|---|---|
| **w/o HPM** | 100 | 81.80 ± 0.14 | 0 | 6.10 |
| **w/ HPM** | 21 | 81.89 ± 0.15 | 0.243 | 1.52 |

**Efficiency Analysis.** The results presented in Table 7 indicate that hard-pair mining effectively filters out 79% of all pairs while requiring only 0.243 ms per few-shot learning task. This leads to a significant decrease in the total run time, from 6.10 ms to just 1.52 ms. This reduction stems from computing distances for only 21% of pairs selected by the hard-pair mining. Additionally, using hard-pair mining even slightly improves performance (81.80% *vs.* 81.89%).

## 5.5 Visualization

We present estimated feature distributions for the mini-ImageNet dataset using horopca for dimensionality reduction. We applied the 1-nearest-neighbor algorithm (with Poincaré distance) to compute classification boundaries, shown in Figure 5. Our method (Figure 5(b)) corrects misclassifications present in Figure 5(a), resulting in more uniform classification zones. Additionally, the distance between query points and prototypes is closer, improving cohesion within categories. For example, in Figure 5(a), yellow and red prototypes are closely positioned, causing yellow query samples to fall into the red zone. Our method (Figure 5(b)) effectively separates these prototypes, correcting the misclassification. By expanding distances between prototypes, our method enhances discriminative capabilities and strengthens class identification by clustering query samples near the new prototypes, leading to clearer class separation. More visualization can be found in the Appendix E.

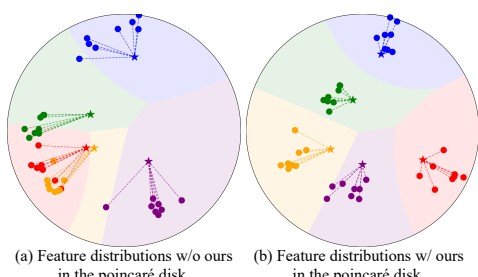

(a) Feature distributions w/o ours in the poincaré disk.   (b) Feature distributions w/ ours in the poincaré disk.

Figure 5: Feature distribution on the mini-ImageNet dataset for 5-ways, 5-shots, and 8-queries. Dotted lines connect prototypes (⋆) and query samples (●). Shaded regions represent classification areas, with colors indicating categories. Comparison is shown for w/o and w/ our method.

## 6 Conclusion

In this paper, we have presented geometry aware hyperbolic distance measures that accommodate diverse hierarchical data structures through adaptive projection matrix and curvature. The adaptive curvature endows embeddings with more flexible hyperbolic spaces that better match the inherent hierarchical structures. The low-rank projection matrices bring positive pairs closer and push negative pairs farther apart. Moreover, the hard-pair mining mechanism enables the efficient selection of hard cases from the query set without introducing additional parameters, reducing the computational cost. Theoretical analysis and experiments show the effectiveness of our method in refining hyperbolic learning through geometry aware distance measures.

**Limitations.** A primary limitation of our work is potential bias, as the distance measure generator may be sensitive to specific data distributions. Additionally, while we use low-rank decomposition to reduce computational costs, this is a preliminary implementation. Future work will explore other decomposition methods to maintain geometric consistency and enhance efficiency.

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

# A POINCARÉ BALL MODEL

Hyperbolic space is a smooth Riemannian manifold with constant negative curvature $c$ and has five isometric models (Beltrami, 1868; Cannon et al., 1997), including Lorentz (hyperboloid) model, the Poincaré ball model, Poincaré half-space model, the Klein model, and the hemisphere model. Here, we consider the Poincaré ball model (Cannon et al., 1997) in light of optimization simplicity and stability. The Poincaré ball model of an $n$-dimensional hyperbolic space with curvature $c(c < 0)$ is defined as a Riemannian manifold $\left(\mathbb{B}_c^n, h_c^B\right)$, where $\mathbb{B}_c^n = \{\boldsymbol{x} \in \mathbb{R}^n : -c\|\boldsymbol{x}\| < 1, c < 0\}$ is the open ball with radius $1/\sqrt{|c|}$. The tangent space at $\boldsymbol{x} \in \mathbb{B}_c^n$, a Euclidean space, is denoted by $T_{\boldsymbol{x}}\mathbb{B}_c^n$. The Riemannian metric $h_c^B$ at $\boldsymbol{x}$ is defined as $h_c^B = {\lambda_{\boldsymbol{x}}^c}^2 h^E$, where $h^E = \boldsymbol{I}$ is the Euclidean metric tensor and the conformal factor $\lambda_{\boldsymbol{x}}^c$ is defined as

$$\lambda_{\boldsymbol{x}}^c := \frac{2}{1 + c\|\boldsymbol{x}\|^2}. \tag{9}$$

We use the Möbius gyrovector space (Ungar, 2001) that provides operations for hyperbolic learning and several used operations are shown as follows.

**Addition.** For a pair $\boldsymbol{x}, \boldsymbol{y} \in \mathbb{B}_c^n$, the Möbius addition is defined

$$\boldsymbol{x} \oplus_c \boldsymbol{y} = \frac{\left(1 - 2c\langle \boldsymbol{x}, \boldsymbol{y}\rangle_2 - c\|\boldsymbol{y}\|^2\right)\boldsymbol{x} + \left(1 + c\|\boldsymbol{x}\|^2\right)\boldsymbol{y}}{1 - 2c\langle \boldsymbol{x}, \boldsymbol{y}\rangle_2 + c^2\|\boldsymbol{x}\|^2\|\boldsymbol{y}\|^2}. \tag{10}$$

**Distance measure.** The geodesic distance between two points $\boldsymbol{x}, \boldsymbol{y} \in \mathbb{B}_c^n$ can be obtained as

$$d_c(\boldsymbol{x}, \boldsymbol{y}) = \frac{2}{\sqrt{c}}\operatorname{arctanh}\left(\sqrt{c}\,\|-\boldsymbol{x} \oplus_c \boldsymbol{y}\|\right). \tag{11}$$

**Exponential map.** The exponential map $\operatorname{expm}_{\boldsymbol{x}}^c(\boldsymbol{v})$ projects a vector $\boldsymbol{v}$ from the tangent space $T_{\boldsymbol{x}}\mathbb{B}_c^n$ to the poincaré ball $\mathbb{B}_c^n$,

$$\operatorname{expm}_{\boldsymbol{x}}^c(\boldsymbol{v}) = \boldsymbol{x} \oplus_c \left(\tanh\left(\sqrt{|c|}\frac{\lambda_{\boldsymbol{x}}^c\|\boldsymbol{v}\|}{2}\right)\frac{\boldsymbol{v}}{\sqrt{|c|}\|\boldsymbol{v}\|}\right). \tag{12}$$

**Logarithmic map.** The logarithmic map $\operatorname{logm}_{\boldsymbol{x}}^c$ maps a vector $\boldsymbol{y} \in \mathbb{B}_c^n$ from the poincaré ball to the tangent space $T_{\boldsymbol{x}}\mathbb{B}_c^n$,

$$\operatorname{logm}_{\boldsymbol{x}}^c(\boldsymbol{y}) = \frac{2}{\sqrt{|c|}\lambda_{\boldsymbol{x}}^c}\operatorname{arctanh}\left(\sqrt{|c|}\,\|-\boldsymbol{x} \oplus_c \boldsymbol{y}\|\right)\frac{-\boldsymbol{x} \oplus_c \boldsymbol{y}}{\|-\boldsymbol{x} \oplus_c \boldsymbol{y}\|}. \tag{13}$$

**Matrix multiplication.** In the Gyrovector space, the Möbius matrix multiplication $\otimes_c$ for matrix $\boldsymbol{M} \in \mathbb{B}$ and vector $\boldsymbol{x} \in \mathbb{B}$ is defined as

$$\boldsymbol{M} \otimes_c \boldsymbol{x} = \frac{1}{\sqrt{|c|}}\tanh(\frac{\|\boldsymbol{M}\boldsymbol{x}\|}{\|\boldsymbol{x}\|}\operatorname{arctanh}(\sqrt{|c|}\|\boldsymbol{x}\|))\frac{\boldsymbol{M}\boldsymbol{x}}{\|\boldsymbol{M}\boldsymbol{x}\|} \tag{14}$$

**Hyperbolic Averaging.** We use Einstein mid-point as the counterpart of Euclidean averaging in hyperbolic space. The Einstein mid-point has the most simple form in Klein model $\mathbb{K}$, thus for $(\boldsymbol{x}_1, \ldots, \boldsymbol{x}_N) \in \mathbb{B}$, we first map $\{\boldsymbol{x}\}$ from $\mathbb{B}$ to $\mathbb{K}$, then do the averaging in Klein model, and finally map the mean in $\mathbb{K}$ back to $\mathbb{B}$ to obtain the poincaré mean:

$$\boldsymbol{u}_i = \frac{2\boldsymbol{x}_i}{1 + c\|\boldsymbol{x}_i\|^2}, \ \ \overline{\boldsymbol{u}} = \frac{\sum_{i=1}^N \gamma_i \boldsymbol{u}_i}{\sum_{i=1}^m \gamma_i}, \ \ \overline{\boldsymbol{x}} = \frac{\overline{\boldsymbol{u}}}{1 + \sqrt{1 - c\|\overline{\boldsymbol{u}}\|^2}}, \tag{15}$$

where $\boldsymbol{u}_i \in \mathbb{K}$, $\overline{\boldsymbol{u}}$ is the mean in $\mathbb{K}$, $\overline{\boldsymbol{x}}$ is the mean in $\mathbb{B}$, and $\gamma_i = \frac{1}{\sqrt{1 - c\|\boldsymbol{x}_i\|^2}}$ is the Lorentz factor.

**$\delta$-hyperbolicity.** The Gromov $\delta$-hyperbolicity (Gromov, 1987) is a measure of how closely the hidden structure of data resembles a hyperbolic space. A lower value of $\delta$ implies that the data

exhibits a higher degree of intrinsic hyperbolic structure. The Gromov $\delta$-hyperbolicity is computed as follows. First, we start from the Gromov product for $\boldsymbol{x}, \boldsymbol{y}, \boldsymbol{z} \in \mathbb{X}$, denoted as

$$(\boldsymbol{y}, \boldsymbol{z})_x = \frac{1}{2}\bigg( d\,(\boldsymbol{x}, \boldsymbol{y}) + d\,(\boldsymbol{x}, \boldsymbol{z}) - d\,(\boldsymbol{y}, \boldsymbol{z}) \bigg). \tag{16}$$

where $\mathbb{X}$ is an arbitrary space endowed with the distance function $d$. Following (Fournier et al., 2015), we compute the pairwise Gromov product of all the data, and the results of all pairs are denoted as a matrix $\boldsymbol{A}$. Then $\delta$-hyperbolicity is computed by

$$\delta = (\max_k \min\{\boldsymbol{A}_{ik}, \boldsymbol{A}_{kj}\}) - \boldsymbol{A}. \tag{17}$$

Relative $\delta$-hyperbolicity is computed by $\delta_{rel} = \frac{2\delta(\mathbb{X})}{\mathrm{diam}(\mathbb{X})} \in [0, 1]$, where $\mathrm{diam}(\mathbb{X})$ denotes the set diameter (maximal pairwise distance). Values of $\delta_{rel}$ closer to 0 indicate a stronger hyperbolicity of a dataset. The value of $\delta_{rel}$ on the image datasets we used are shown in Table 8. As can be seen from Table 8, these image datasets all have a clear hierarchical structure (the $\delta_{rel}$ of these datasets is close to 0).

Table 8: The relative delta $\delta_{rel}$ values calculated for different datasets. For image datasets, we measured the Euclidean distance between the features produced by our feature extractors. Values of $\delta_{rel}$ closer to 0 indicate a stronger hyperbolicity of a dataset. Results are averaged across 1000 subsamples of size 20000.

| Dataset | Encoder | $\delta_{rel}$ |
|---|---|---|
| CIFAR10 | Wide-Res 28×2(Zagoruyko & Komodakis, 2016) | 0.354 |
| CIFAR100 | Wide-Res 28×2(Zagoruyko & Komodakis, 2016) | 0.280 |
| Mini-ImageNet | ResNet-12(He et al., 2016) | 0.328 |
| Tiered-ImageNet | ResNet-12(He et al., 2016) | 0.228 |

# B MORE ANALYSES OF OUR OBSERVATION ON THE HYPERBOLIC SPACE VOLUME AND GEODESIC DISTANCE.

## B.1 THE RELATIONSHIP BETWEEN THE CURVATURE AND SPACE VOLUME.

Research in differential geometry has unveiled a profound relationship between curvature and spatial volume across various geometric structures. Topping et al. (Topping, 2008) demonstrated that the volume of a sphere increases in proportion to its mean curvature for a given diameter, elucidating a fundamental principle encapsulated in Theorem B.1. Additionally, Wu et al.(Wu & Zheng, 2011) provided further insights by establishing that higher curvature enables the accommodation of more intricate structures, such as submanifolds, within a given space.

**Theorem B.1.** *Consider an $n$-dimensional hyperbolic manifold $\mathbb{B}_c^n = \mathbf{x} \in \mathbb{R}^n : -c||\mathbf{x}|| < 1, c < 0$. There exists a constant $D(n)$, dependent solely on $n$, such that the intrinsic diameter $d_{int}$ of the hyperbolic manifold and its curvature $c$ are related by the inequality $d_{int} \leq D(n) \int_{\mathbb{B}_c^n} |c|^{n-1} d\mu$.*

This theorem sheds light on the intrinsic connection between the diameter of a hyperbolic manifold and the distribution of its curvature, emphasizing the influence of curvature on the spatial extent of such geometries.

## B.2 THE RELATIONSHIP BETWEEN GEODESIC AND GEOMETRY COMPLEXITY.

**Definition B.2** (Complexity of the hierarchical structures for the pair in hyperbolic space)**.** The $\mathbf{o}$ is the origin of the n-dimensional hyperbolic manifold $\mathbb{B}_c^n = \mathbf{x} \in \mathbb{R}^\mathbf{n} : -\mathbf{c}||\mathbf{x}|| < \mathbf{1}, \mathbf{c} < \mathbf{0}$ . The complexity of the hierarchical structure between $x_1, x_2$ is designated by

$$C(x_1, x_2) = P(x_1 \to o) + P(x_2 \to o),$$

where $P(\cdot \to o)$ is the connectivity from $x$ to $o$, measured by the connected graph distance(Balbuena et al., 1996).

The distance measure (determined by curvature $c$) and defined complexity are related in the following intuitive way. We explain it in the 2-dimensional case where the hierarchical structure is represented by a tree, the simplest planar graph. Now for any two vertices $x_1$ and $x_2$ in the tree, the complexity between them is just the number of edges through which the path between them passes, or in other words, the number of hierarchical levels their path runs across. For pairs of different complexity, the learned curvatures and geodesics are supposed to lie in different "phase spaces". If one insists on picturing them as a common space (which might turn out to be what the neural network actually configures), the geodesic edges hence the hierarchical structure would better be embedded into a deformed hyperbolic surface with sufficiently large genus (to allow for enough homotopy classes of the geodesics i.e. larger fundamental group) rather than as a planar tree. Then the total complexity or total curvature of the hierarchical structure is afforded by the topological complexity of the deformed hyperbolic surface, i.e. its genus or equivalently its Euler characteristics. Note that for a (closed orientable) hyperbolic surface $S$ of genus $g$, its Euler characteristic $\chi(S) = 2 - 2g$. Actually, the total curvature of a compact closed surface $S$ equals to $2\pi\chi(S)$ by the classical Gauss-Bonnet theorem. More generally, total curvature and Euler characteristic are related by Cohn-Vossen's inequality.

**Theorem B.3** (Cohn-Vossen's inequality(Cohn-Vossen, 1959))**.** *For a non-compact complete surface $S$ without boundary and $K$ its Gaussian curvature,*

$$\int_S K dS \leq 2\pi\chi(S).$$

Thus we see for surfaces bearing hierarchical structures, to allow more complexity or equivalently more topological complexity, i.e. large genus (more negative Euler characteristics), $|K|$ and $c$ must be varied according to the complexity of the structure. In higher dimensions, the explanation of curvature variation is not as intuitive as in the surface case, but it seems to be a similar consequence of topological rigidity since Mostow rigidity theorem says the geometry of higher dimensional (complete finite-volume) hyperbolic manifolds are determined by their fundamental groups(Mostow, 1968).

Besides the topological reasoning above, the connected paths between pairs of points in a hierarchical structure (i.e., trees) are designated to be expressed by truly geometric geodesics between pairs in hyperbolic manifolds(Masur & Minsky, 1998). In the context of this paper, a conformal relationship between geodesics and complexity is sought. The existence of such geodesics is guaranteed by the Hopf-Rinow theorem in complete Riemannian manifolds (see chapter 7, Theorem 2.8 of (Do Carmo & Flaherty Francis, 1992)). Moreover, it can be generalized to locally compact path connected metric geometry, the conditions of which are automatically satisfied by the spaces in our consideration, as follows.

**Theorem B.4** (Hopf-Rinow theorem(Gromov et al., 1999))**.** *If $(X, d)$ is a complete, locally compact path metric space, then:*
*1. Closed balls are compact, or, equivalently, each bounded, closed domain is compact.*
*2. Each pair of points can be joined by a minimizing geodesic.*

In our case, the geodesic distance between $x_1, x_2$ are given by Equation (2) :

$$d(x_1, x_2) = \frac{2}{\sqrt{c}} \operatorname{arctanh}\left(\sqrt{c} \left| -x_1 \oplus_c x_2 \right|\right).$$

Equation (2) demonstrates that with increased curvature, geodesics become steeper with a larger distance. Higher curvature corresponds to "steeper" geodesics $d(x_1, x_2)$ (larger distance), which can conform to complex hierarchical structures with higher $C(x_1, x_2)$, as shown in Figure 1.

## C    MODE DETAILS ABOUT CURVATURE GENERATOR

Our design prioritizes computational efficiency and embeds specific inductive biases (e.g., the bilinear form and bounding the curvature) to capture the geometry of hyperbolic space effectively. We choose the factorized bilinear pooling yu2017multi to produce suitable curvature by using expressive second-order information of data. Second-order information unveils the dynamics and trends within data, akin to how curvature reflects the degree of warping in the hyperbolic space. Using this information allows us to comprehend the local curvature of data distributions, deepening our grasp of the geometric

structures. Gao et al. (2021) have empirically demonstrated the effectiveness of utilizing second-order information to generate curvature. Details about the factorized bilinear pooling are shown as follows.

**Details about the factorized bilinear pooling.** We use a factorized bilinear pooling that produces suitable curvature by using expressive second-order information of data, where the sum_pooling and sigmoid function are used to reduce dimension of second-order information and constrain the produced curvature $|c|$ in a valid range.

In a non-factorized bilinear pooling method, the second-order information of a pair of data ($x_1 \in \mathbb{R}^n$, $x_2 \in \mathbb{R}^n$) is the outer products $x_1 x_2^T$, and the curvature $|c|$ can be computed by a fully-connected layer $f(\cdot)$, $|c| = f(x_1 x_2^T)$. The second-order information captures expressive characterization of data, and thus can produce suitable curvature. However, the non-factorized bilinear pooling method has large computation consumption, since the dimension of $x_1 x_2^T \in \mathbb{R}^{n \times n}$ is high, causing a large number of parameters in $f(\cdot)$. To solve this issue, the parameter of $f(\cdot)$ is factorized into two parameters via matrix factorization, and the two parameters are applied to two fully-connected layers $f_1(\cdot)$ and $f_2(\cdot)$, respectively.

In this case, $f(x_1 x_2^T)$ can be rewritten as $f(x_1 x_2^T) = \mathbb{1}(f_1(x_1) \circ f_2(x_2))$, where $\mathbb{1}$ is an all one-vector, serving as a summation function (sum_pooling). The detailed derivation can be found in (Yu et al., 2017). The range of $\mathbb{1}(f_1(x_1) \circ f_2(x_2))$ is $(-\infty, \infty)$, while some hyperbolic methods (Khrulkov et al., 2020; Ermolov et al., 2022) show that the curvature $|c|$ in a valid range(eg. $[0,1]$) is suitable for much data. Thus, we use a sigmoid function $|c| = sigmoid(sum\_pooling(f_1(x_1) \circ f_2(x_2)))$ to rescale it into a valid range.

# D   PROOF OF THEOREM 4.2

In this section, we will present a theoretical analysis of low-rank approximation mainly based on polynomial partitioning which is strengthened the polynomial method in incidence geometry, and Talagrand concentration inequality (Talagrand, 1995) which is now a fundamental tool in random matrix theory. We provide the upper bound of the error (as shown in Eq. (18)) in the gyro-vector multiplication, as well as the lower bound of the probability for this upper bound to hold. Without specialization, the vector $x$ is bounded away from the boundary, *i.e.* $\|x\| \leq c < 1$ for some constant $c > 0$ and the range of data $Mx$ is bounded.

## D.1   MOTIVATION AND VALIDATION OF USING RANDOM MATRIX THEORY

The low-rank approximation is a commonly used technique in computation to reduce computational complexity and cost. Its effectiveness is usually attributed to the following Eckart-Young-Mirsky theorem:

**Theorem D.1** ((Eckart & Young, 1936; Mirsky, 1960)). *Let $\| \cdot \|_2$ be spectral norm on $M \in \mathbb{R}^{m \times n}$. Suppose $A \in M$ has singular value decomposition $A = \sum_{j=1}^{r} \sigma_j u_j v_j^T$ with $\sigma_1 \geq \sigma_2 \geq \cdots \sigma_r \geq 0$. If $k \geq r$, then the matrix $A_k = \sum_{j=1}^{k} \sigma_j u_j v_j^T$ satisfies*

$$\|A - A_k\|_2 \leq \|A - B\|_2, \text{ for any } B \in M \text{ with rank at most } k.$$

So if we use a rank-$k$ matrix to approximate a target matrix with a rapidly decaying spectrum, the effectiveness is already easily validated by the above theorem. For example in cluster analysis, if a graph consists of $n - k$ sub-graphs which are weakly connected to each other, its eigenvalues $\sigma_i, i \geq k + 1$ would be close to zero whence has good rank-$k$ approximation. However, whether the matrices in practice have a rapidly decaying spectrum is a random issue. More precisely, the matrices in consideration may vary according to some distribution, typically such as Gaussian i.i.d or so, which only allows us to investigate how well the matrices can be approximated by low-rank ones in the sense of probability. Though this is a natural question in machine learning, solid theoretical analysis has been rarely done on this matter to our knowledge. Moreover, the matrices in our consideration act by Möbius multiplication, which requires more careful specialized analysis through a meshing process as we will show below.

## D.2 Approximation with Möbius multiplication

For short, we denote $M = M^{res} \in \mathbb{R}^{n \times n}$ and $M' = M_a^{res} M_b^{res\top} \in \mathbb{R}^{n \times n}$, where $M_a^{res} \in \mathbb{R}^{n \times k}$ and $M_b^{res} \in \mathbb{R}^{n \times k}$ so that $M_a^{res} M_b^{res\top}$ is of rank $\leq k \ll n$ (say $k < n/10$). Then simply ($\| \cdot \|$ denotes vector norm)

$$
\begin{aligned}
error &= \|(I + M) \otimes x - (I + M') \otimes x\| \\
&= \|M \otimes x - M' \otimes x\|.
\end{aligned}
\tag{18}
$$

Recalling from the gyro-matrix multiplication formula, we have for any $M \in \mathbb{R}^{n \times n}, x \in \mathbb{B}^n$ ($\|x\| < 1$):

$$
M \otimes x = \underbrace{\tanh\left(\frac{\|Mx\|}{\|x\|} \operatorname{arctanh}(\|x\|)\right)}_{a} \underbrace{\frac{Mx}{\|Mx\|}}_{b}.
\tag{19}
$$

Denote the corresponding terms of $M' \otimes x$ by $a'$ and $b'$. Since $-1 < \tanh(x) < 1$, we know $|a|, |a'| < 1$. Note also that $\|b\| = \|b'\| = 1$. Then simply by the triangle inequality,

$$
\begin{aligned}
\|M \otimes x - M' \otimes x\| = \|ab - a'b'\| &= \|(a - a')b + a'(b - b')\| \\
&\leq |a - a'|\|b\| + |a'|\|b - b'\| \\
&\leq |a - a'| + \|b - b'\|.
\end{aligned}
\tag{20}
$$

We estimate $|a - a'|$ and $\|b - b'\|$ separately, both of which are far from trivial as in the Euclidean multiplication case (say may be handled by Eckart-Young-Mirsky theorem or so).
(i) For the former $|a - a'|$, there needs a scalar approximation of the hyperbolic trigonometric functions. Let $u = \operatorname{arctanh}(\|x\|) = \frac{1}{2} \ln\left(\frac{1+\|x\|}{1-\|x\|}\right) > 0$ (noticing that $\tanh(x) = \frac{e^x - e^{-x}}{e^x + e^{-x}}$ and $\|x\| \leq c < 1$), $\delta = \frac{\|Mx\|}{\|x\|}$ and $\delta' = \frac{\|M'x\|}{\|x\|}$. Since the function $\tanh'(x) = \frac{1}{\cosh^2(x)} \leq 1$, by Lagrange's mean value theorem we get

$$
|a - a'| \leq u|\delta - \delta'|.
\tag{21}
$$

Note that $u$ is bounded since $\|x\| \leq c < 1$. Hence we need only concern about $|\delta - \delta'|$, the difference between Rayleigh quotients, which can not be handled simply by SVD and Eckart-Young-Mirsky theorem D.1 or so. Suppose $x = \sum_{j=1}^{n} a_j v_j$. Then $Mx = \sum_{j=1}^{n} \sigma_j a_j u_j$, while $M'x$ is supposed to converge to $\sum_{j=1}^{k} \sigma_j a_j u_j$. By Eckart-Young-Mirsky theorem D.1, to better approxmate $M = \sum \sigma_j u_j v_j^T$ under SVD, the generator should converge to $M_a^{res} = \sum_{j=1}^{k} \mu_j u_j$ and $M_b^{res} = \sum_{j=1}^{k} \nu_j v_j$ with $\mu_j \nu_j = \sigma_j$. But we do not know which direction is the major contribution to $\delta = \frac{\|Mx\|}{\|x\|}$ and set-up $\delta'$ may drift away. More precisely, recall the classical min-max theorem:

**Theorem D.2** (see (Hwang, 2004)). *For any integer $m \geq 1$, let $S_m$ be set of all sub-spaces of dimension $m$ of $\mathbb{R}^n$. Then*

$$
\min_{V \in S_{n-m+1}} \left\{ \max_{x \in V \setminus \{0\}} \frac{\langle Mx, x \rangle}{\|x\|^2} \right\} = \sigma_m,
\tag{22}
$$

*where the minimum is attained for $V$ orthogonal to $u_j$'s (or $v_j$'s) appearing in SVD of $M$, $j = 1, \ldots, m-1$.*

This simply tells us that learning the Rayleigh quotient $\delta$ for $x$ lying in different sub-spaces would result in totally different low-rank matrices and even any averaging on learned matrices seems irrational.
(ii) For the latter $\|b - b'\|$, it is important to note that $b = \frac{Mx}{\|Mx\|}$ represents vectors projected onto the unit sphere $S^{n-1}$. This requires an analysis of randomized linear regression on higher dimensional spheres as introduced in the next subsection.

As we will explain, the real learning process through a multi-layered network should subdivide the whole space of rank-$k$ matrices into local meshes according to data distribution and store a learned

matrix for each local mesh, i.e. it is not a real matrix but a piece-wise linear operator varying on meshes. The effectiveness of learning on both the scalar approximation and linear regression on spheres as in (i) and (ii) is an implication of Talagrand concentration inequality, which is fundamental in random matrix theory.

### D.3 MESHING FEATURE SPACES BY POLYNOMIAL PARTITIONING AND TALAGRAND CONCENTRATION INEQUALITY

Recalling that the matrix generator $g_t()$ as in (5) of section 4.1 is a local generator which appears to generate a matrix for each pair. In actual learning stage, it is not likely to fluctuate drastically pairwise but rather stays invariant amongst pairs within a group of similar features, i.e. it only varies as a piecewise linear transformation. Thus our primary concern lies with samples of $M^{res}x_i$, where $x_i$'s are the training data sampled from the feature space of pictures or so. When sampled finely enough, these samples may be effectively considered as distributed within a local mesh of data, akin to a Gaussian distribution with a small variance, as demonstrated in subsequent analyses.

We first introduce the material technique of "divide and conquer" for sampling general sets through polynomial partitioning, developed by Larry Guth and Nets Katz when dealing with the Erdős' distinct distances problem (L. Guth, 2015).

**Theorem D.3** (Theorem 4.1 of L. Guth (2015)). *For any set $S \subset \mathbb{R}^n$ of $N$ points and positive integer $d$, there exists a hypersurface $Z$ defined by a polynomial of degree $\leq c2^{d/n}$ for some absolute constant $c > 0$, whose complement $\mathbb{R}^n \smallsetminus Z$ is the union of $2^d$ disjoint open cells each containing $\leq 2^{-d}N$ points of $S$.*

The key point here is that we need very few partitioning parameters (determining a suitable polynomial) to segment big data sets into much smaller clusters with similarities as desired especially if the data distributes relatively continuously. Note that the hypersurface itself may contain part of the points. If the samples distribute relatively continuously, we may see these surface points locally as residing on a tangent space of the hypersurface, which is of dimension $n-1$. Hence we may deal with the surface points inductively so that we need only concern about the points in cells. For example, if we require each cell to have $n$ points, i.e. $d \sim \log(N/n)$, we only need $O(2^{\log(N/n)/n}) = O(\sqrt[n]{N/n})$ parameters. In practice, if we use training sets about size in millions and feature dimension in hundreds, the partitioning parameters are freely spared without any concern.

The above explained polynomial partitioning mechanism seems to reveal at least partially the dark matter of miracle efficiency of even shallow-layered neural networks. From a more general theoretical standpoint, any neural network is designed to use elementary operations to segment and model global datasets with local subsets that align with ambient distributions in assumption, which naturally calls for and results in polynomial partitioning outcomes. This particularly facilitates the application of low-rank approximations.

A baby version of low-rank approximation in Euclidean case may be tried out by an older and simpler result in topology, called *Ham sandwich theorem* which is a version of the Borsuk-Ulam theorem, to deal with approximation of local pieces after partitioning.

**Theorem D.4** (Theorem 4.2 of Guth-Katz (L. Guth, 2015)). *Any $n$ open sets of finite volume in $\mathbb{R}^n$ can be simultaneously bisected by a single hyperplane.*

Thus if say each cell contains $n-1$ feature points after polynomial partitioning, which we may allow to reside in small neighborhoods bounding errors, there is a hyperplane bisecting the neighborhoods and passing the auxiliary origin point 0. Suppose the hyperplane is defined by $a^T x = 0$ for some $a \in \mathbb{R}^n$. Then the matrix $I + aa^T$ (with $aa^T$ working as $M^{res}$) is the suitable approximation.

Of course, a more sophisticated method is needed to deal with the general case especially when data encounters randomized noises. In our context, the network seeks a substantially lower-dimensional subspace, specifically the range space of $M_a^{res}M_b^{res\top}x_i$, to control $|\delta - \delta'|$ as in (i) and $\|b - b'\|$ as in (ii) while projected onto $S^{n-1}$. This projection acts as a linear regression for samples within each subset. We will first deal with (ii) and identify an optimal $b'$ that minimizes $\|b - b'\|$, a process we refer to as *randomized linear regression*.

The feasibility of randomized linear regression is underpinned by a key theoretical element: the *Talagrand concentration inequality* (Talagrand, 1995), which provides the foundational argument for the possibility of effective randomized linear regression in our context.

**Theorem D.5** ( see Corollary 2.1.19 of Tao (Tao, 2023)). *Let $X \in \mathbb{R}^n$ ($\mathbb{C}^n$) be a random vector with entries of independent random variables with mean 0 and variance 1, and bounded almost surely by $K$. Let $V$ be a subspace of $\mathbb{R}^n$ ($\mathbb{C}^n$) of dimension $k$. Then for any $\lambda > 0$, one has*

$$\mathrm{Prob}(|d(X, V) - \sqrt{n-k}| \geq \lambda K) \leq Ce^{-c\lambda^2}$$

*for some absolute constants $C, c > 0$.*

### D.4 ESTIMATE OF (II) $\|b - b'\|$

Now we choose a subspace $V$ of dimension $k$, which contains the mean $\mu$ of $X$ (now ranging locally in a suitable mesh of $M^{res}x_i$). Note that in practice, we may assume that $k \leq n/10$ (or $k \leq \sqrt{n}$) and the number of training samples $N$ is large (say $\sim 100n$). One key idea of applying the above Talagrand concentration inequality is that, the space $V$ is supposed to be the range of approximating low rank ($\leq k$) matrices, hence determines our choices of $M'$. Another key idea is that according to Theorem A.3 in Guth-Katz (L. Guth, 2015) , we can use a low-degree ($O(\sqrt[n]{N/k})$) hypersurface to segment out $k$ feature points into each cell of its complement. Then our model is supposed to construct a subspace $V$ of dimension $k$ to approximate the points in each cell and only allow systematic error due to data collecting, feature embedding or so. Assuming the entrywise variance of error of $X$ to be $\sigma^2$ so that $\frac{X-\mu}{\sigma}$ has entrywise variance 1, then the theorem shows

$$\begin{aligned}
\mathrm{Prob}\left(\left|d(\frac{X-\mu}{\sigma}, V) - \sqrt{n-k}\right| \geq \lambda K\right) &= \mathrm{Prob}\left(\left|d(X - \mu, V) - \sigma\sqrt{n-k}\right| \geq \lambda K\sigma\right) \\
&= \mathrm{Prob}\left(\left|d(X, \mu + V) - \sigma\sqrt{n-k}\right| \geq \lambda K\sigma\right) \\
&= \mathrm{Prob}\left(\left|d(X, V) - \sigma\sqrt{n-k}\right| \geq \lambda K\sigma\right) \\
&\leq Ce^{-c\lambda^2},
\end{aligned} \tag{23}$$

where $\mathrm{Prob}(\cdot)$ denotes the probability. Note that the last equality holds since $\mu \in V$ and the dimensions $k$ and $n$ are relatively fixed. When projected onto the sphere $S^{n-1}$, the distance $d(X, V)$ is rescaled to be $\sim \arcsin\frac{d(X,V)}{\|\mu\|}$, since distance on unit sphere is measured by the central angle $\theta$ with $\sin\theta = \frac{d(X,V)}{\|\mu\|}$, as indicated as in Figure 6. We assume that the variance of systematic error is $\sigma = n^{-1/2}\epsilon$, for $\epsilon > 0$ sufficiently smaller than the mean $\|\mu\|$, so that $K = m\|\mu\|$ with $m \sim 1$ (since $X$ has small variance in assumption). To make $\lambda K\sigma$ significantly smaller than $\sigma\sqrt{n} = \epsilon$, we choose $\lambda = \sqrt{k/c}$ so that $\lambda K\sigma = \sqrt{\frac{k}{cn}}m\|\mu\|\epsilon \ll \epsilon$ noting that $k \ll n$ and $m \sim 1, \|\mu\| < 1$. Then by (6), with high probability $(1 - Ce^{-k})$, $d(X, V)$ is concentrated around $\sigma\sqrt{n} = \epsilon$ and the angle $\theta$ below is small since

$$\arcsin\frac{d(X, V)}{\|\mu\|} \sim \frac{d(X, V)}{\|\mu\|} \sim \frac{\epsilon}{\|\mu\|} \ll 1.$$

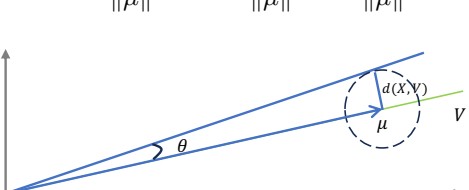

Figure 6: $X$ distributed around the mean $\mu$ which lies in the subspace $V$.

To be clear, by $\sim$ we mean bounded both below and above by bounded factors close to 1. For most of the subsets of a suitable meshing of $M^{res}x_i$, the magnitude of the mean $1 > \|\mu\|$ is sufficiently bounded away from 0. Thus we have abundant choices of $V$, hence $M^{res}$, for randomized linear regression. Note that the samples are meshed in dimension $k$, i.e. rank-$k$ approximation is used. Altogether the above argument shows the following

**Lemma D.6.** *Suppose the variance of systematic error $\sigma^2 = \epsilon^2/n$ and $\epsilon$ is significantly smaller than (say $\sim \frac{1}{10}$ of) the mean of error, and the samples feature space distribute relatively continuously. Then with high probability ($1 - Ce^{-k}$ for some constant $C > 0$ say for $k \sim n/10$ or $k \sim \sqrt{n}$), for the estimate of (ii) we have*

$$\|b - b'\| < \epsilon, \tag{24}$$

*for $b' = \frac{M'x}{\|M'x\|}$ with some matrix $M'$ of rank $\leq k$ (say $\leq n/10$ or $\sim \sqrt{n}$).*

Thus for rank $k$ smaller than $n$ but not too small (say $n$ in hundreds while $k$ in dozens work well as shown by experiments), we have plenty of choices of $M'$ of rank $\leq k$ to well approximate the target matrix $M$ in local meshes.

Next, we deal with the estimate of (i) $|a - a'| = u|\delta - \delta'|$ which further puts restrictions on the choices of sub-spaces $V$.

## D.5    ESTIMATE OF (I) $|a - a'| = u|\delta - \delta'|$

Note that we assume $\|x\| \leq c < 1$ so that $u = \operatorname{arctanh}(\|x\|) \leq C_1$ is bounded for some $C_1 > 0$. Hence we only need to bound $|\delta - \delta'|$. More explicitly,

$$
\begin{aligned}
|\delta - \delta'| &= \frac{\big|\|Mx\| - \|M'x\|\big|}{\|x\|} = \frac{\big|\|Mx\|^2 - \|M'x\|^2\big|}{\|x\| (\|Mx\| + \|M'x\|)} \\
&= \frac{|\langle (M - M')x, Mx \rangle + \langle (M - M')x, M'x \rangle|}{\|x\| (\|Mx\| + \|M'x\|)}.
\end{aligned} \tag{25}
$$

The denominator of (25) may be written $\|x\|^2 \left( \frac{\|Mx\|}{\|x\|} + \frac{\|M'x\|}{\|x\|} \right)$, in which the first Rayleigh quotient is supposed to be bounded away from $0$ to define a well deformed metric and so is the second as an approximation (say close to the form as in Eckart-Young-Mirsky theorem D.1). Also, there is no harm to assume $\|x\| = 1$, i.e. restricting the Rayleigh quotients onto the sphere $S^{n-1}$. Hence the denominator of (25) is also bounded by some constant $C_2 > 0$.

Thus we only need to deal with the numerator $N = |\langle (M - M')x, Mx \rangle + \langle (M - M')x, M'x \rangle|$ of (25). Again we choose any $k$-dimensional sub-space $V$ passing the mean of data $X$ as the range space of $M'x$. Let $\pi_V$ be the projection of $\mathbb{R}^n$ onto $V$. Note that $I_n - \pi_V$ is the projection onto the orthogonal space of $V$. A specialization of $M' = \pi_V M$ definitely gives an upper bound for the numerator, i.e.

$$
\begin{aligned}
\text{best estimate of } N &\leq |\langle (M - \pi_V M)x, Mx \rangle + \langle (M - \pi_V M)x, \pi_V Mx \rangle| \\
&= \langle (M - \pi_V M)x, Mx \rangle \\
&= \langle (M - \pi_V M)x, (M - \pi_V M)x \rangle \\
&= \|(M - \pi_V M)x\|^2, \tag{26}
\end{aligned}
$$

the last term of which is actually a scale of the distance $d(Mx, V)$ on the sphere $S^{n-1}$ as shown in Figure 6. The scaling is bounded by the length of $Mx$, which as we assumed in Talagrand inequality (Theorem D.5) is bounded by a constant $K > 0$. Thus by (26) and lemma D.6, we can estimate the numerator of (25) as

**Lemma D.7.** *Following the same assumptions and notations with lemma D.6, with high probability ($1 - Ce^{-k}$ for some constant $C > 0$), for the estimate of (i) we have*

$$|a - a'| = u|\delta - \delta'| < \frac{KC_1}{C_2}\epsilon, \tag{27}$$

*for $K$ bounding the range of data $X$, $C_1, C_2 > 0$ and $M'$ of rank $\leq k$.*

Finally, altogether by lemma D.6 and D.7, we conclude the Theorem 4.2.

# E    MORE ABLATION AND VISUALIZATION

## E.1    EXPLOITATION OF SYMMETRY

Our distance measure cannot ensure the strict symmetry of input pair $(x, y)$, since we do not shared the weights of the subnetworks($f_a$ and $f_b$ in the projection matrix $g_t$, $f_1$ and $f_2$ in the curvature generator $g_c$) within the generator, where $f_a$ and $f_1$ are used for $x$, and $f_b$ and $f_2$ are used for $y$. But we do not argue this is a problem. For a sample pair $(x, y)$ in the training process, we randomly sample $x$ and $y$ from the dataset, and the probability of $(x, y)$ and $(y, x)$ are equal. Thus, if we change $(x, y)$ to $(y, x)$, and send it to our distance measure, their distances will not change significantly.

To demonstrate this point, we do the ablation by swapping $f_a$ and $f_b$ in the projection matrix generator($g_t$) as well as $f_1$ and $f_2$ in the curvature generator($g_c$), which is equal to swapping the inputs of $g_t$ and $g_c$ separately. As shown in Table 9, the impact of swapping the subnetworks in the projection matrix generator and the curvature generator on the performance of our method is negligible. This indicates that $f_a$ and $f_b$ , as well as $f_1$ and $f_2$ , have learned nearly identical knowledge. Although our method does not ensure the symmetry, it avoids situations where swapping $x$ and $y$ in a pair leads to a sharp change in distance, showing the robustness of our method.

Table 9: Ablations about the symmetry. 5-shot accuracy(%) and and 95 % confidence interval on the mini-ImageNet and tiered-ImageNet.

|  | mini-ImageNet | tiered-ImageNet |
|---|---|---|
| **Ours** | $81.80 \pm 0.14$ | $85.22 \pm 0.16$ |
| **Swapping $f_a$ and $f_b$** | $81.81 \pm 0.14$ | $85.20 \pm 0.16$ |
| **Swapping $f_1$ and $f_2$** | $81.80 \pm 0.14$ | $85.22 \pm 0.16$ |

## E.2    EXPLOITATION OF THE DIFFERENT LOSS FUNCTIONS.

We also have tried the supervised contrast loss function, specifically,

$$L(x_i, x_j, x_k) = \max(0, margin + dis(x_i, x_j) - dis(x_i, x_k))$$

, where $x_i$ is the anchor sample, $x_j$ is the positive sample, and $x_k$ is the negative sample. Our algorithm is capable of converging with this type of loss function as well as shown in Table 10. However, since it equally penalizes all negative samples without considering their hierarchical differences, it may hinder the model's ability to differentiate between less relevant and highly irrelevant samples. Ideally, this loss function would require the true hierarchical structure as ground truth, but such detailed annotations are absent in the current visual datasets. Consequently, we chose direct cross-entropy loss

$$L = -\sum_{i=1}^{C} y_i \log(p_i)$$

, where $y_i$ is the label and $p_i$ is the probability of the prediction, allowing the network to implicitly learn the diverse geometric structures among different pairs.

Table 10: Ablations about the symmetry. 5-shot accuracy(%) and and 95 % confidence interval contrastive loss vs. cross-entropy loss on the mini-imagenet dataset.

|  | 5-ways 1-shot | 5-ways 5-shot |
|---|---|---|
| **Contrastive loss** | $61.91 \pm 0.20$ | $79.45 \pm 0.14$ |
| **Cross-entropy Loss** | $64.75 \pm 0.20$ | $81.89 \pm 0.15$ |

## E.3    EXPLOITATION OF THE STRUCTURE OF THE PROJECTION MATRIX GENERATOR.

In the design of the projection matrix generator, we use two separate low-rank residual matrices $M_a^{res} M_b^{res\top}$ to approximate to the projection matrix. We also test the results of using a unified low-rank residual matrix $M_a^{res} M_a^{res\top}$. Results in Table 11 show that asymmetrical designs outperform symmetrical ones in effectiveness. Symmetrical designs result in symmetric projection matrices,

which are limited to linear transformations and may not capture the nonlinearity of real-world scenes. In contrast, our proposed geometry-aware distance measure leverages an asymmetric structure, allowing for a more adaptable and diverse fit.

Table 11: Accuracy(%) of symmetric design vs. ours on CIFAR 10 dataset and CIFAR 100 dataset.

|  | Symmetric form($M_a^{res}M_a^{res\top}$) | Ours ($M_a^{res}M_b^{res\top}$) |
|---|---|---|
| **CIFAR10** | 88.64 | 96.56 |
| **CIFAR100** | 71.02 | 75.61 |

### E.4 RESULTS OF STANDARD CLASSIFICATION ON MULTIPLE BACKBONES

The experimental results on WideRes-28, ResNet-50, and ResNet-101 using the CIFAR100 dataset are summarized in Table 12. Our method demonstrates consistent improvements across all tested backbones, achieving an accuracy increase of 1.78% on WideRes28-2, 2.7% on ResNet50, and 2.48% on ResNet101. These results underscore the effectiveness of our proposed adaptive hyperbolic distance measure in enhancing model performance, regardless of the architecture, and confirm its generalizability to diverse network structures.

Table 12: Accuracy (%) on WideRes-28, ResNet-50 and ResNet-101 on the CIFAR100 datasets.

| Backbone | Fixed | Ours |
|---|---|---|
| WideRes28-2 | 73.83 | 75.61 |
| Resnet50 | 78.49 | 81.19 |
| Resnet101 | 80.97 | 83.45 |

### E.5 COMPARISON WITH HIERARCHICAL-AWARE PROTOTYPE POSITIONING

We conducted experiments comparing our method with hierarchical-aware prototype positioning (Ghadimi Atigh et al., 2021). Specifically, we replaced the backbone in (Ghadimi Atigh et al., 2021) with the same backbone (Wide ResNet-28-2) used in our method, and set the dimension to 128, consistent with our approach. The results shown in Table 13 indicate that our method still exhibits significant superiority. We will add these results in the revised version.

Table 13: Comparison with Hyperbolic Bussman Learning (Ghadimi Atigh et al., 2021) on CIFAR 10 and CIFAR 100 dataset. * means implemented by ourselves.

| Method | CIFAR 10 (%) | CIFAR 100 (%) |
|---|---|---|
| HBL* | 91.16 | 67.42 |
| Ours | 94.75 | 75.61 |

### E.6 HARD CASES

We visually compare the logits before and after applying our method to hard cases. We calculate the probabilities by applying the softmax function to the logits, and the results are shown in Figure 7. Each cell in Figure 7 represents a query sample's probabilities of 5-ways task, where each bar denotes the probability of being classified into a specific category, with the red bar indicating the correct category. Within each cell of the Figure 7 , the left figure represents the probabilities without our method, and the right figure demonstrates that using our method. It can be seen that, after applying our method to hard cases, the probability of the correct class increases significantly, indicating that our method is able to correct misclassification caused by fixed distance measures.

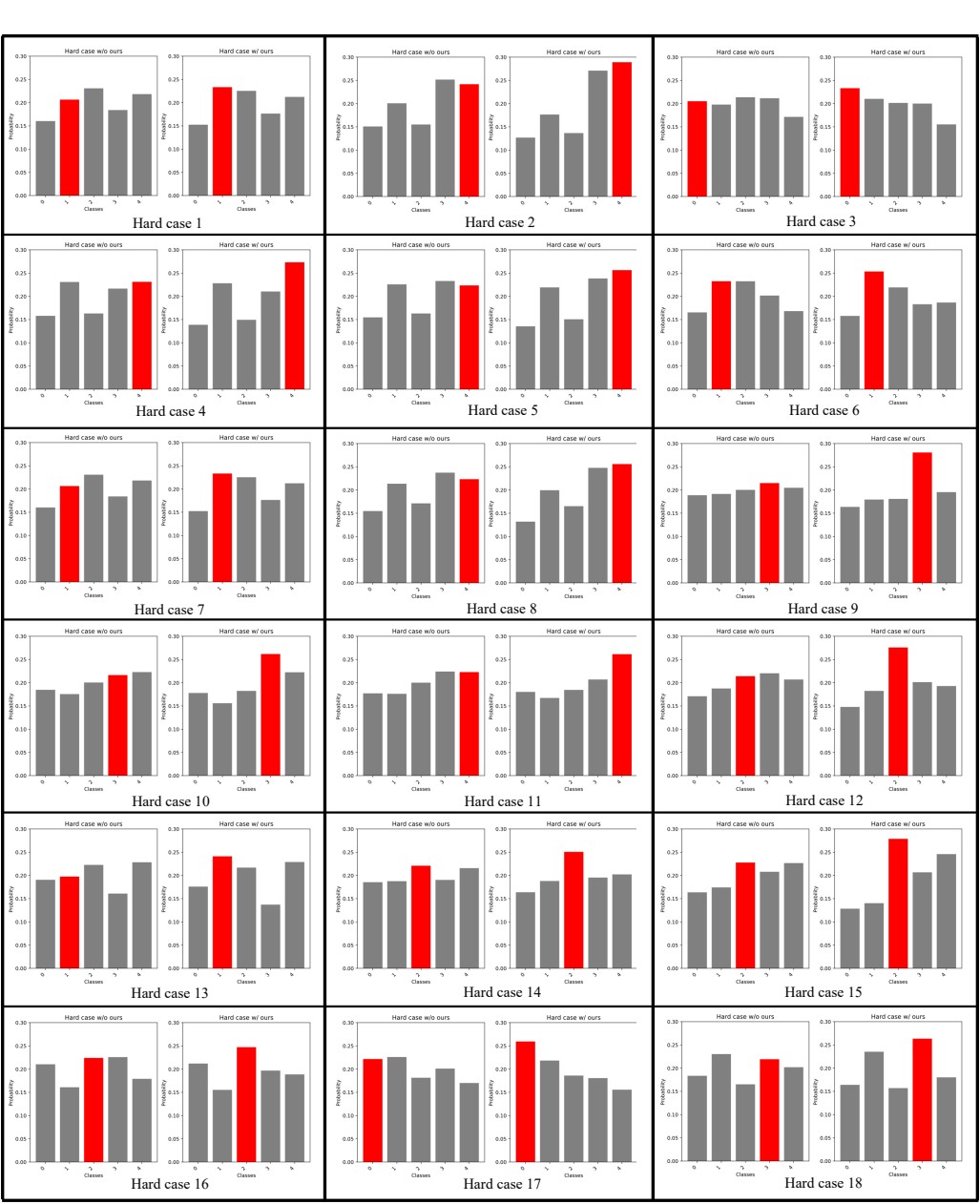

Figure 7: Probability distributions of several hard cases on 5 ways, 5 shots task of the mini-ImageNet dataset. In each table cell, the histogram on the left shows the class probability distribution without using our method, while the one on the right shows the distribution with our method applied. The horizontal axis represents classes, and the vertical axis represents probability. The red bar represents the correct class.

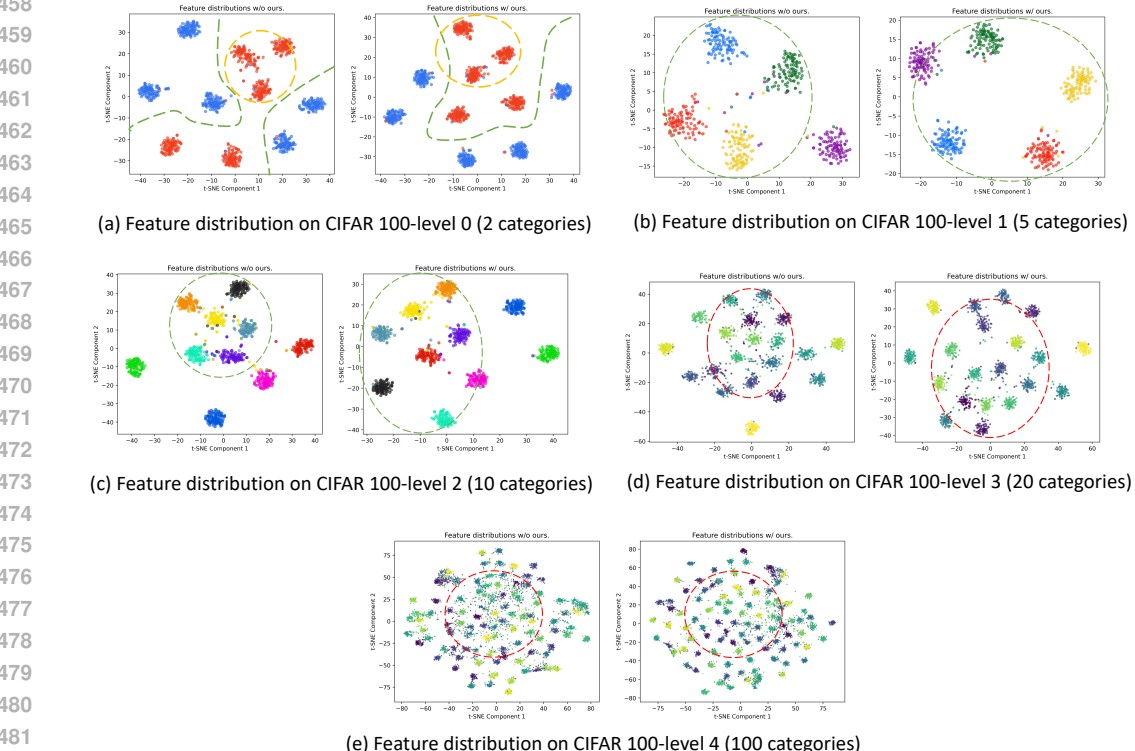

(a) Feature distribution on CIFAR 100-level 0 (2 categories)

(b) Feature distribution on CIFAR 100-level 1 (5 categories)

(c) Feature distribution on CIFAR 100-level 2 (10 categories)

(d) Feature distribution on CIFAR 100-level 3 (20 categories)

(e) Feature distribution on CIFAR 100-level 4 (100 categories)

Figure 8: Feature distribution of the 5-level annotations on the CIFAR 100 dataset via T-SNE. For levels 1-4, we randomly sampled one fine category (out of 100) from each coarse category. For level 0, we sampled five from each.

### E.7 DISTRIBUTION VISUALIZATION

#### E.7.1 HIERARCHICAL EMBEDDING DISTRIBUTION

We visualize the distribution of our hyperbolic embeddings at the 5 hierarchical levels, as shown in Figure 8. We observe that our method leads to clearer boundaries among different categories on the 5 hierarchical levels, showing that our method can better capture hierarchical structures again.

#### E.7.2 CURVATURE DISTRIBUTION AND CASE STUDY

We visualize the curvature distributions in the middle epoch and the final epoch when training on the CIFAR-10 and CIFAR-100 datasets, as shown in Figure 9. We observe that the training process in CIFAR-10 pushes the curvatures to small values, while the training process in CIFAR-100 pushes the curvatures to big values. The reason is that the CIFAR-10 dataset has a relatively simple hierarchical structure, while CIFAR-100 has a complex hierarchical structure. This is also confirmed by the delta hyperbolicity values in Table 8 in the **Appendix** A, where CIFAR-100 has a smaller delta hyperbolicity value than CIFAR-10, showing the more complex hierarchical structure in CIFAR-100.

#### E.7.3 VISUALIZATION VIA HOROPCA

We draw the feature distributions in the mini-ImageNet dataset using the horopca(Chami et al., 2021) dimensionality reduction methods. We also plot the classification zones with different colors. The classification zones are computed by 1-nearest-neighbor algorithm, using the poincaré distance. Results are shown in Figure 10, 11. Our visualization analysis demonstrates that our method effectively corrects misclassifications observed in the left subfigures, leading to more averaged classification zones. This improvement results in a clearer separation between different classes,

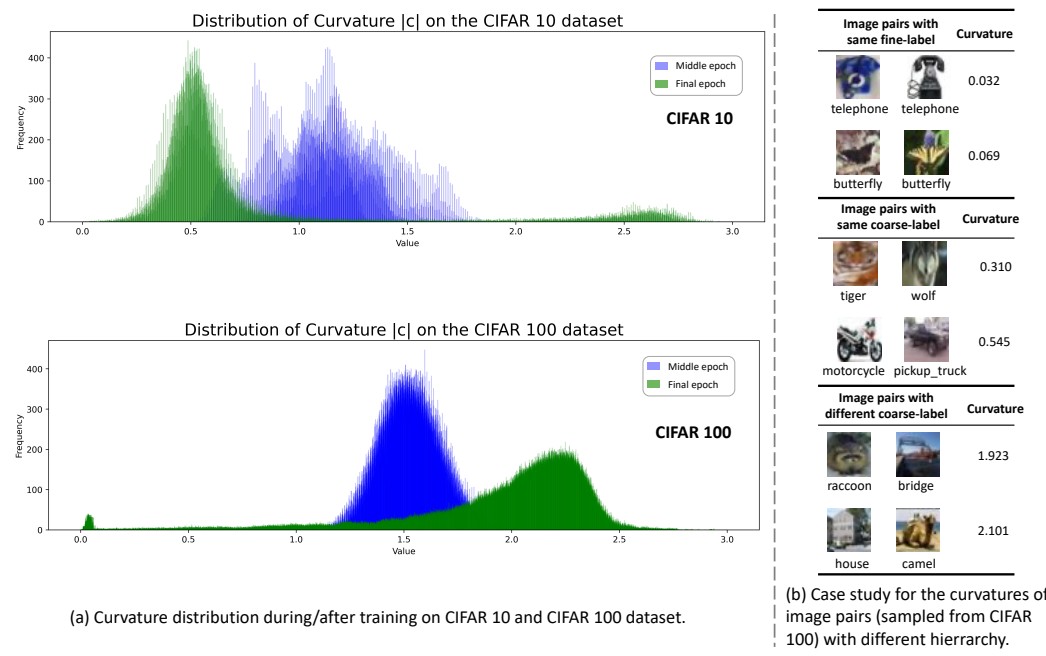

(a) Curvature distribution during/after training on CIFAR 10 and CIFAR 100 dataset.

(b) Case study for the curvatures of image pairs (sampled from CIFAR 100) with different hierrarchy.

Figure 9: Curvature distribution during (middle epoch) and after (final epoch) training and case study. The range of the learned curvature is set to [0.0001, 3.0] in the experiment of (a). In (b), image pairs with closer levels have lower curvature $|c|$, while image pairs with greater hierarchical differences have higher curvature.

enhancing the model's precision. Furthermore, our approach brings query points significantly closer to their respective class prototypes, achieving stronger intra-class cohesion. This indicates a more compact clustering of data points within the same category, which directly contributes to the model's improved classification accuracy. Our method not only addresses and rectifies errors in class assignment but also promotes a more organized and interpretable representation of classes through tighter clustering and clearer boundaries. This advancement underscores our model's capability to deliver superior classification performance with enhanced reliability and specificity.

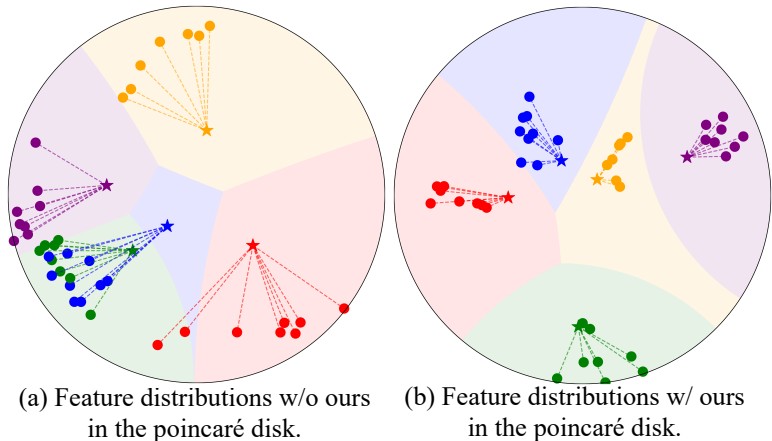

(a) Feature distributions w/o ours in the poincaré disk.

(b) Feature distributions w/ ours in the poincaré disk.

Figure 10: An example of feature distribution (w/o ours *vs*. w/ ours) on the mini-ImageNet dataset for the setting of 5-ways, 5-shots, and 8-queries. Different colors indicate different categories. ⋆ indicates the prototype. ● indicates the query sample.

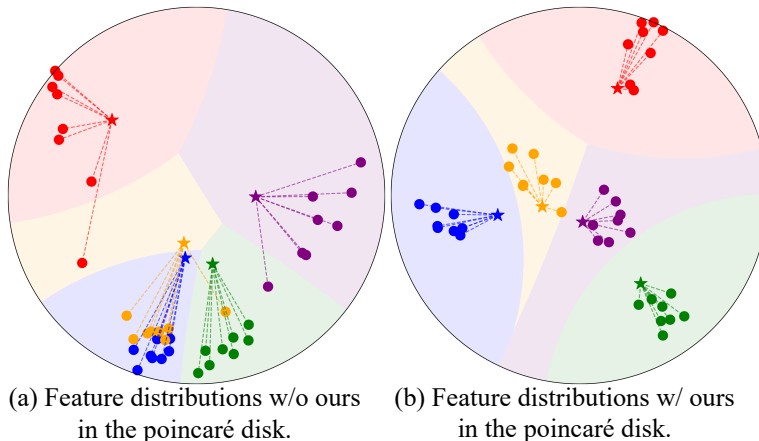

(a) Feature distributions w/o ours
in the poincaré disk.

(b) Feature distributions w/ ours
in the poincaré disk.

Figure 11: An example of feature distribution (w/o ours *vs.* w/ ours) on the mini-ImageNet dataset for the setting of 5-ways, 5-shots, and 8-queries. Different colors indicate different categories. ⋆ indicates the prototype. • indicates the query sample.

### E.7.4 VISUALIZATION VIA T-SNE

We also draw the feature distributions in the mini-ImageNet dataset using the t-SNE dimensionality reduction methods. Results are shown in Figure 12, 13, 14. From the figures, we have the following conclusions: (1) Our method increases the distance between prototypes and improves the discriminative ability of prototypes. (2) By making the query samples closely surround the new prototypes, the method enhances the ability to discern and distinguish different classes.

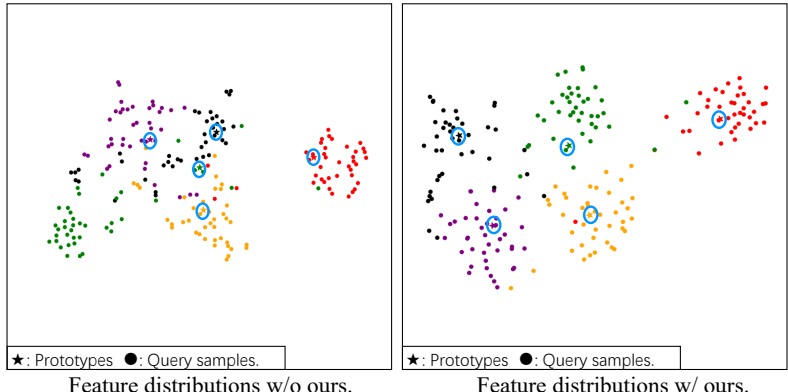

Feature distributions w/o ours.

Feature distributions w/ ours.

Figure 12: An example of feature distribution (w/o ours *vs.* w/ ours) on the mini-ImageNet dataset for the setting of 5-ways, 5-shots, and 40-queries. Different colors indicate different categories. ⋆ indicates the prototype, highlighted by the blue circle. • indicates the query sample.

## F MORE EXPERIMENTAL DETAILS

### F.1 STANDARD CLASSIFICATION

#### F.1.1 DATASET

We conduct experiments on three popular datasets, namely MNIST(LeCun & Cortes, 2010), CIFAR10(Krizhevsky et al., 2009), and CIFAR100(Krizhevsky et al., 2009). MNIST contains 10 classes with 60000 training images and 10000 testing images. Each image has a resolution of $28 \times 28$, and the numerical pixel values are in greyscale. The CIFAR-10 dataset consists of 60000 color images in

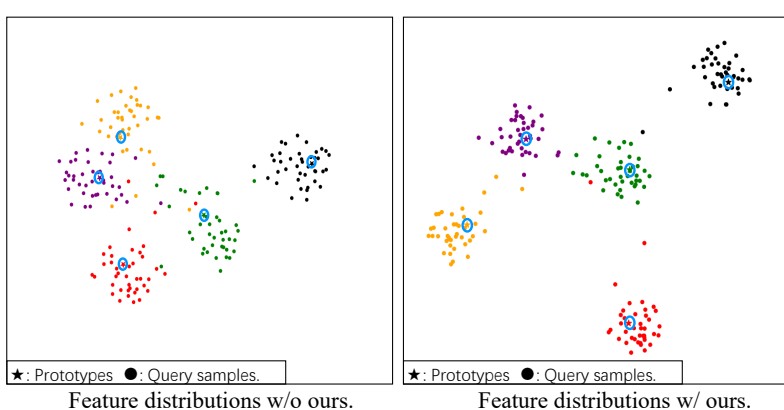

Feature distributions w/o ours.  Feature distributions w/ ours.

Figure 13: An example of feature distribution (w/o ours *vs.* w/ ours) on the mini-ImageNet dataset for the setting of 5-ways, 5-shots, and 40-queries. Different colors indicate different categories. ⋆ indicates the prototype, highlighted by the blue circle. • indicates the query sample.

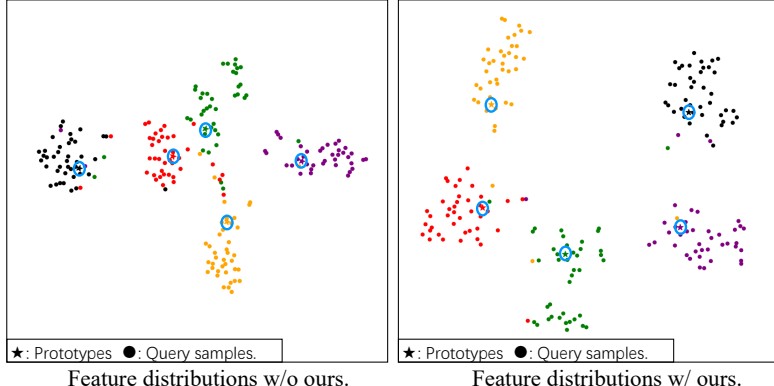

Feature distributions w/o ours.  Feature distributions w/ ours.

Figure 14: An example of feature distribution (w/o ours *vs.* w/ ours) on the mini-ImageNet dataset for the setting of 5-ways, 5-shots, and 40-queries. Different colors indicate different categories. ⋆ indicates the prototype, highlighted by the blue circle. • indicates the query sample.

10 classes, each class having 6000 images with a size of $32 \times 32$. We use 50000 images for training and 10000 images for testing. CIFAR-100 has 100 classes containing 600 $32 \times 32$ color images each, with 500 training and 100 testing images per class.

### F.1.2 MODEL DETAILS

Our method regards that the features from the backbones are located in the hyperbolic space's tangent space at the origin, and we utilize an exponential map on top of the backbone to transfer these features from the tangent space to the hyperbolic space. We pre-train the backbones on the training set using the cross-entropy loss. The backbones are fixed in our metric learning process. We calculate Einstein mid-point of all the features from the training set as the prototypes and do the classification according to the similarities between samples and prototypes. For the MNIST(LeCun & Cortes, 2010) dataset, we use a LeNet-like(LeCun et al., 1998) net as the backbone network. For CIFAR10(Krizhevsky et al., 2009) and CIFAR100(Krizhevsky et al., 2009) we use Wide-ResNet $28 \times 2$(Zagoruyko & Komodakis, 2016) as the backbone network. The Wide-ResNet $28 \times 2$ is trained for 120 epochs with the SGD optimizer. The learning rate is set to $0.01$ at first and decayed per 30 epochs with a decay rate of $0.1$.

### F.2 HIERARCHICAL CLASSIFICATION

### F.2.1 DATASET

Using the CIFAR100 dataset and its 5-level hierarchical annotations from (Wang et al., 2023), we compared our adaptive distance measure with a fixed one, using Resnet50 and Resnet101. The category IDs of the 5 hierarchical levels are as follows:

- Level 4 (100 categories): The original fine labels in CIFAR-100.

- Level 3 (20 categories): The original coarse labels in CIFAR-100.

- Level 2 to Level 0: Constructed based on the 20 coarse labels (Level 3), provided by [a]. Specifically:

  - Level 2: ([0-1]), ([2-17]), ([3-4]), ([5-6]), ([12-16]), ([8-11]), ([14-15]), ([9-10]), ([7-13]), ([18-19]).
  - Level 1: ([0-1-12-16]), ([2-17-3-4]), ([5-6-9-10]), ([8-11-18-19]), ([7-13-14-15]).
  - Level 0: ([0-1-7-8-11-12-13-14-15-16]) and ([2-3-4-5-6-9-10-17-18-19]).

### F.3 FEW-SHOT LEARNING

### F.3.1 DATASET

We conduct the few-shot learning task on mini-ImageNet (Vinyals et al., 2016) and tiered-ImageNet (Ren et al., 2018) datasets. The mini-ImageNet dataset contains 100 classes from the ImageNet dataset(Russakovsky et al., 2015), containing 600 images for each class. We split the 100 classes into 64, 16, and 20 classes for training, validation, and testing, respectively. The tiered-ImageNet dataset has 779165 images from 608 classes, where 351, 97, and 160 classes are used for training, validation, and testing, respectively. All images in both mini-ImageNet and tiered-ImageNet datasets are resized to $84 \times 84$.

### F.3.2 MODEL DETAILS

We use the ResNet-12(He et al., 2016) plus the exponential map as the feature extractor. We pre-train the ResNet-12 on the training set over 120 epochs with the SGD optimizer, using the cross-entropy loss. In the pre-training stage, the learning rate is initially set to $0.01$. The learning rate decays by $0.1$ after every 40 epochs. Once the pre-training is completed, we remove the last fully-connected layer and the softmax layer of the pre-trained model, and the rest layers are used in our feature extractor. The feature extractor is fixed in the learning process of our distance measures generator.

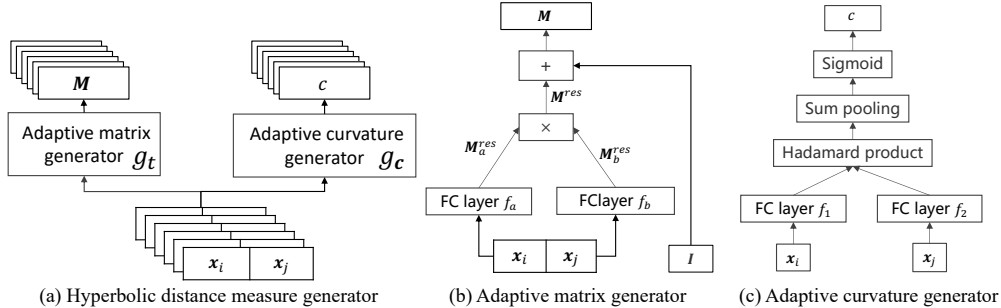

Figure 15: Architecture of the hyperbolic distance measure generator. $\boldsymbol{x}$ are the feature points.

## F.4 EXPERIMENTAL CONFIGURATION

We use an Intel(R) Xeon(R) Gold 6226R 2.90GHz CPU, a GeForce RTX 3090 GPU, and 256GB RAM to conduct experiments. We use CUDA 12.0, Python 3.8.12, and PyTorch 1.13.1.

## F.5 GENERATOR DETAILS

As illustrated in Fig. 15, our distance measure generator consists of a projection matrix generator and a curvature generator. For a pair of data points $\boldsymbol{x}_i, \boldsymbol{x}_j$, the curvature generator produces a curvature that accommodates their hierarchical structure. Subsequently, the projection matrix generator produces two low-rank matrices $M_a^{res}, M_b^{res}$, which, with the support of a residual strategy, complete the generation of the projection matrix.

