# OpenReview forum: "Geometry-aware Distance Measure for Diverse Hierarchical Structures in Hyperbolic Spaces"
_ICLR.cc/2025/Conference — ICLR 2025 Conference Withdrawn Submission_

### Official Review · Reviewer_fXFZ · 2024-10-21

**Soundness:** 3
**Presentation:** 3
**Contribution:** 3
**Rating:** 5
**Confidence:** 3

**Summary:**

The paper introduces a novel approach to learning adaptive geometry-aware distance measures in hyperbolic spaces. The paper proposes learning adaptive projection and curvature for each pair of data points, allowing distance measures to dynamically adjust to the hierarchical relationships between samples. To address the computational complexity of generating unique projections and curvatures for each data pair, the authors introduce a low-rank decomposition scheme to reduce complexity and a hard-pair mining mechanism that focuses only on challenging sample pairs. The authors provide theoretical analysis showing that the low-rank approximation yields small errors with high probability. Experiments demonstrate the effectiveness of the proposed method.

**Strengths:**

Overall quality is good, and the writing is clear. I appreciate the author providing extensive experiments to demonstrate the effectiveness of their methodology, which adds to the soundness of their approach. The novelty is fine for the topic of learning hyperbolic embeddings. The low rank approximation theoretical analysis will be a novel framework in this field.

**Weaknesses:**

- The motivation of some operations is questionable:
  - Why is learning projection matrix M_ij necessary when from Fig. 1, the main difference between (c) and (d) is the curvature? Could you provide a more detailed justification?
  - How to compare with other methods with curvature c as a single learnable parameter? How to compare with other methods with curvature c as a tunable hyperparameter? I suggest authors include these comparisons in the experimental section (or feel free to point these out if I missed them) and provide discussions for the comparison of these methodologies.
  - What is the motivation of using M^res? Why will M^res mitigate instability of training? Can you provide any intuition?
  - The motivation of using different curvatures might be due to the artifact that classes are placed at the boundary of the space as pictured in Fig. 1. In Fig. 1 (b) why can’t we shift the dog-wolf pair up, closer to the origin of the space? Why not rescale the weight of the subtrees (for example, scale up the dog-wolf pair) so that you can embed the full tree? I’m asking because there exists combinatorial algorithms [1] to embed trees into Poincare disk with arbitrarily low distortion.


[1] Rik Sarkar. Low Distortion Delaunay Embedding of Trees in Hyperbolic Plane. In Proceedings of the 19th International Conference on Graph Drawing, GD’11, pages 355–366, Berlin, Heidelberg, 2012. Springer-Verlag.

- Writing of the proof of Prop. 4.1:
  - Since this is the core motivation of using adaptive curvature, please consider adding some proof sketch/intuition in the main body, and provide more explanation for terminologies for the proof in the Appendix.
  - line 212: Could you define “better conform to” formally?
  - Please add the citation for Theorem B.1 in its statement in line 956-957, like the format you use for Theorem B.4.
  - It would be better to explain terminologies in Sec B.2 for readers without extensive background in hyperbolic geometry. Readers might be familiar with operations in line 162-176, but not something like “deformed hyperbolic surface”, “genus”, “homotopy classes”, “fundamental group” etc., from line 972-986.

I would love to raise my score if you can address all my concerns and questions below.

**Questions:**

- Theorem 4.2: For readability, please consider elaborating on “systematic error”, “mean of error”, “relatively continuously” in the main section. Or, to save some space, you can write an informal version of Theorem 4.2, for example, removing the range of constants in the statement.
- Line 300: Either move Algorithm 1 to Page 6 or explain how you compute the “prototype” (by Einstein mid-point based on the Algorithm, but now it’s after where you first mentioned prototype). There can be different ways for prototype computation, not limited to your current approach, or Frechet mean, so it would be better to clarify this.
- Line 333, when you update g_t and g_c, do you keep the feature extracted by ResNet fixed? How do you choose the optimizer between SGD and Riemannian SGD because your final feature data points are in the hyperbolic space?
- Figure 5: how do you visualize this figure if each pair uses different curvatures? What does this disk boundary stand for? Is it this standard 2-d Poincare disk with curvature = -1?
- Figure 12: Why do you use TSNE if feature embeddings are non-Euclidean? TSNE implicitly assumes Euclidean distance metrics for the local similarity/Gaussian computation, maybe Hyperbolic UMAP is a better option?

---

### Official Review · Reviewer_bjxg · 2024-11-01

**Soundness:** 3
**Presentation:** 3
**Contribution:** 3
**Rating:** 6
**Confidence:** 3

**Summary:**

This work presents a dynamically adjustable geometric distance measurement method designed to accommodate various hierarchical structures in hyperbolic space. The model can learn the projections and curvature of samples and map them to an appropriate hyperbolic space. Additionally, the use of low-rank decomposition methods and hard-pair mining allows the model to reduce the computational burden associated with pairwise generation without compromising accuracy.

**Strengths:**

1.Dynamic learning of various hierarchical structures in hyperbolic space is the core innovation of this work.
2.The article is logically coherent, with rigorous expression and solid theoretical derivation.

**Weaknesses:**

1.There are several inaccuracies in the mathematical expressions throughout the article that require careful review and correction by the authors.
2.How was the hierarchical structure of the dataset used in the experiments generated?
3.In Table 3, the absence of the latest comparison algorithms from this year diminishes the persuasive power of the experiments.

**Questions:**

See Weaknesses

---

### Official Review · Reviewer_poaT · 2024-11-02

**Soundness:** 2
**Presentation:** 1
**Contribution:** 2
**Rating:** 3
**Confidence:** 4

**Summary:**

The Authors have proposed a learning algorithm using geometry-aware distance measure determined by the matrix generator and curvature generator.

**Strengths:**

1. The motivation of the paper, considering the diversity of the hierarchical structures of data, is nice.
2. As far as I know, the proposed model is novel.

**Weaknesses:**

Overall, despite the interesting starting motivations, the current manuscript fails to include the necessary definitions for readers to understand the Authors' idea. Each of the following points is, regrettably, fatal and also prevents us from judging the validity of other parts, such as experiment parts since, in general, experiments are conducted to verify whether the Authors idea is correct or not.
1. The specific forms of the matrix generator $g\_{t}$ and curvature generator $g\_{c}$ are not given in the paper. One might say it is a general form, but leaving these as a general form is unacceptable. If we do not restrict the function class of $g\_{t}$ and $g\_{c}$, we can learn any distance structure for the feature pairs as $\\boldsymbol{M}\_{ij}$ can depend on $ij$. This means the proposed method can overfit any data infinitely, and we can no longer say the model considers specific representation space, e.g., hyperbolic space.
2. Many mathematical terms are used in Theorem 4.2 without definitions. For example,

- systematic error
- relatively continuously
- absolute value calculation to define $|\\cdot|$ (for a vector, like $\\boldsymbol{M \otimes x} - \\boldsymbol{M' \otimes x} $)

are not defined. Also, I could not see "the mean of error" of *what* in Line 264. To be honest, I am unsure whether this theorem is correct since I do not see any trick in Theorem's setting that makes the low-rank matrix $\\boldsymbol{M}'$ approach $\\boldsymbol{M}$ or the feature vectors bound in the subspace of the kernel (linear-algebraic sense) of $\\boldsymbol{M}' - \\boldsymbol{M}$. In any case, we cannot verify it without specific definitions of those terms given by the Authors.

**Questions:**

See Weaknesses.

---

### Official Review · Reviewer_QWUf · 2024-11-10

**Soundness:** 2
**Presentation:** 2
**Contribution:** 2
**Rating:** 5
**Confidence:** 5

**Summary:**

## Summary

The paper proposes a geometry-aware methodology for learning hierarchical structures in hyperbolic geometry. It attempts to address the limitations of previous algorithms for hyperbolic learning where the hierarchical structures are assumed to be uniform and lead to distortion in the learnt hierarchies. The proposed algorithm includes a geometry-aware adaptive distance measurement where the distance adjusts based on the hierarchical complexity by using pairwise low-rank projection and adjustment of the curvature. Experimental results on classification and few-shot learning tasks on 3 datasets are provided.

**Strengths:**

## Strengths

1. The key motivation of the paper to adaptively learn distance metrics for different hierarchies is an interesting and relevant problem for the hyperbolic learning community.


2. The proposed solution is novel and intuitive.


3. Theoretical analysis for the method is provided using the Talagrand concentration inequality which is helpful for further analysis in the field.


4. Visualizations on different datasets are helpful in understanding the effectiveness of the learning algorithm.

**Weaknesses:**

## Weaknesses

1. The authors mention that their work is inspired by the Gu et. al, 2019 paper in section 2.3 (L141-147) - Learning mixed curvature representations in Products of Model Spaces -  the nature of the work is similar in terms of learning adaptive curvatures however the authors do not provide a comparison with this method in their experiments



2. There are several details about the experimental setup and comparisons which are unclear from the main paper and the Appendix. The classification accuracy measurement is mentioned as “similarities between samples and prototypes” (L1685), which seems to be different from the standard classification setups used in literature where the accuracies are measured by the softmax logits of the outputs from the classifier layer or k-NN accuracy for classification. Since the features are projected to the hyperbolic space, is the similarity computed as inverse hyperbolic distances to the prototypes or using a different method? What are the backbones and loss function used for experiments in Table 1 and did the authors experiment with Hyperbolic backbones? Additionally, how do the results for the experiments in Table 1 compare with euclidean/non-hyperbolic methods with equivalent setups?


4. While the authors mention the complexity analysis for low rank decomposition, what are the training times for the different settings and how much of an increase does the proposed methodology lead to in terms of actual training time for the experiments?

**Questions:**

## Questions:



1. The paper uses the Poincare model of geometry which is the most common, how do the authors envision their method adapting to other models of hyperbolic geometry, such as the lorentz model which are shown to be more stable [1]  from an algorithmic perspective? Additionally, will the theoretical results still hold?

[1] The Numerical Stability of Hyperbolic Representation Learning, Mishne et. al, ICML '23

---

### Note · Authors · 2024-11-15

I have read and agree with the venue's withdrawal policy on behalf of myself and my co-authors.